# Combinatorial effects of multiple genes contribute to beneficial aneuploidy phenotypes

M Sophie Koller[1,2,3], Claudia Himmelbauer[1,2], Sarah Fink[1,2], Madhwesh C Ravichandran[1,2] & Christopher S Campbell [ID][1,2 ✉]

## Abstract

Aneuploidy is one of the most common adaptive mechanisms to environmental selection in cells, yet its advantages over other genomic alterations remain unclear. We used budding yeast to determine if beneficial aneuploidy phenotypes are driven primarily by combinatorial expression changes of multiple genes. To determine the impact of complex aneuploidy on cellular fitness and drug resistance, we generated yeast collections with nearly every combination of two chromosome gains or losses. Additionally, we genetically dissected aneuploid chromosomes using partial deletions to identify chromosomal regions contributing to aneuploidy-driven drug resistance. Strong resistance phenotypes consistently came from combinations of chromosomes or chromosomal regions, with up to five regions on a chromosome contributing to resistance to a single drug. Gene ontology terms had limited predictive power in identifying the genes contributing to resistance phenotypes, as the combinatorial effects from different aneuploid regions act through multiple resistance pathways. The strongest phenotypes came from synergistic effects between copy number changes of different chromosomes or chromosomal regions, demonstrating how subtle gene expression changes of many genes combine to greatly impact cell survival.

**Subject Category** Chromatin, Transcription & Genomics

## Introduction

The accurate distribution of chromosomes during cell division is essential for maintaining genomic integrity across all eukaryotic species. Chromosome segregation errors result in alterations in chromosome copy number, which are termed aneuploidy. Aneuploidy is one of the most common genetic alterations in cancer, occurring in ~90% of solid tumors and ~70% of hematopoietic cancers (Weaver and Cleveland, 2006). These high frequencies of aneuploidy are not only inherent to cancer cells but are also commonly observed within other vegetatively growing cell populations, including industrial strains of baker's yeast and drug-resistant strains of the human pathogens *Candida albicans* and *Leishmania* (Selmecki et al, 2006; Ubeda et al, 2008; Peter et al, 2018; Loegler et al, 2024).

Despite the frequency of aneuploidy in mitotically growing cell populations, it is generally associated with reduced cellular fitness. Engineering aneuploidy in cells invariably decreases growth rates under optimal conditions (Torres et al, 2007; Stingele et al, 2012; Hintzen et al, 2022; Zerbib et al, 2024). This decrease in fitness results from changes in gene copy numbers of many transcribed genes on the aneuploid chromosome (Bonney et al, 2015). For both yeast and human cells, gene expression levels of lab-engineered aneuploidies scale with gene copy numbers for most of the aneuploid genes (Torres et al, 2007; Stingele et al, 2012; Wangsa et al, 2019; Rojas et al, 2024; Muenzner et al, 2024). These small changes in expression levels for hundreds of genes cause a wide range of deleterious phenotypes. Such general aneuploidy-associated phenotypes include increased proteotoxic stress, cell cycle delays, and increased DNA damage (Torres et al, 2007; Sheltzer et al, 2011; Thorburn et al, 2013; Oromendia et al, 2012; Beach et al, 2017; Li and Zhu, 2022; Williams et al, 2008; Passerini et al, 2016; Santaguida et al, 2015; Garribba et al, 2023).

Yet, aneuploidy is one of the most frequently observed mechanisms of cellular adaptation. The first direct link between adaptive drug resistance and aneuploidy was established in the human pathogen *Candida albicans* (Selmecki et al, 2006). Adaptation via aneuploidy has since been more extensively observed in the budding yeast *Saccharomyces cerevisiae*, where specific aneuploidies consistently emerge during experimental evolution screens under adverse environmental conditions (Gresham et al, 2008; Yona et al, 2012; Chen et al, 2012; Voordeckers et al, 2015; Zhang et al, 2016; Beaupere et al, 2018; Morard et al, 2019). Moreover, inducing aneuploidy via random chromosome missegregation in both yeast and human cells promotes resistance to a large variety of drugs (Pavelka et al, 2010; Chen et al, 2012; Lukow et al, 2021; Ippolito et al, 2021). Aneuploidy can also act as a suppressor of mutations that decrease cellular fitness (Rancati et al, 2008; Liu et al, 2015; Ryu et al, 2016; Ravichandran et al, 2018; Macaluso et al, 2025). In cancer, karyotype patterns have predictive power for drug resistance, tumor progression, and patient outcome (Shukla et al, 2020).

[1]Max Perutz Labs, Vienna Biocenter, Vienna, Austria. [2]University of Vienna, Vienna, Austria. [3]Vienna Biocenter PhD Program, a Doctoral School of the University of Vienna and the Medical University of Vienna, Vienna, Austria. ✉E-mail: christopher.campbell@univie.ac.at

The potential for aneuploidy as an adaptation mechanism to a large variety of selective environments can partially explain the high prevalence of aneuploidy following long-term vegetative growth, such as in cancer or in yeast used for alcohol production (Peter et al, 2018). However, it still fails to explain why cells adapt through aneuploidy over other types of mutations that don't share its inherent downsides. Different models for the specific advantage of adaptation through aneuploidy have been proposed. Aneuploidy has been suggested to be a rapid, but transient solution to sudden environmental changes (Pompei and Cosentino Lagomarsino, 2023). In *S. cerevisiae* subjected to acute heat stress, chromosome 3 is gained early in the adaptation process but then lost again over time (Yona et al, 2012). Similarly, yeast adapting to high rates of chromosomal instability (CIN) adapted first through the accumulation of specific aneuploidies, followed later by the accumulation of suppressor point mutations (Clarke et al, 2023). However, there are also many examples of specific aneuploidies that are maintained throughout long-term adaptation, highlighting that speed is likely not the only advantage of adaptation via aneuploidy (Selmecki et al, 2009; Ravichandran et al, 2018; Adell et al, 2023; Clarke et al, 2023; Girish et al, 2023; Zhou et al, 2024; Watson et al, 2024; Hintzen et al, 2024; Bökenkamp et al, 2025).

An additional explanation for the advantages of aneuploidy as a mechanism of adaptation comes from its ability to change gene expression levels of several beneficial genes simultaneously. In yeast adaptation screens, only a handful of examples could attribute the selective advantage of aneuploidy to changes in the expression levels of one or two specific genes. Even in these cases, they rarely recapitulated the strength of the aneuploidy phenotype (Selmecki et al, 2009; Chen et al, 2012; Ryu et al, 2016; Linder et al, 2017; Beaupere et al, 2018; Ravichandran et al, 2018; Barney et al, 2021; Pavani et al, 2021; Zhou et al, 2024). In cancer, overexpression of individual genes also fails to fully recapitulate the phenotypes of aneuploid chromosomes (Trakala et al, 2021; Su et al, 2021; Girish et al, 2023). These findings suggest that the challenges of identifying driver genes on aneuploid chromosomes could be amplified by the interplay between multiple genes. It is currently unclear how many genes on aneuploid chromosomes typically contribute to resistance to selective pressures.

Analyses of aneuploidy patterns in cancer genomes as well as cellular adaptation experiments have shown strong correlations between specific chromosome or chromosome arm combinations, indicating karyotype selection is influenced by genetic interactions between aneuploid chromosomes (Ozery-Flato et al, 2011; Ravichandran et al, 2018; Shukla et al, 2020; Adell et al, 2023). One such interaction has been observed in budding yeast, where gaining an additional copy of chromosome 6 is lethal, but can be rescued with the additional gain of chromosome 13 (Torres et al, 2007). Whether such genetic interactions between chromosomes are widespread and if they are affected by selective conditions is currently unknown.

In this study, we generated a large yeast collection with nearly every possible combination of two chromosome gains or losses to determine the frequency and strength of genetic interactions between aneuploid chromosomes and how combinatorial aneuploidy phenotypes impact drug resistance and sensitivity. We found that genetic interactions between chromosomes are rare in the absence of treatment but become much more frequent under selective pressure. In addition, the strongest resistance phenotypes

consistently came from combinations of aneuploid chromosomes, demonstrating the advantage of copy number changes in multiple genes. We genetically dissected resistant aneuploid chromosomes using partial arm deletions to determine the number of regions that contribute to aneuploidy-driven drug resistance on a single chromosome. We found that resistance conferred by single disomies nearly always comes from multiple chromosomal regions. While the majority of phenotypes are additive, the strongest resistance phenotypes of aneuploidies came from synergistic effects between different regions or chromosomes.

# Results

## Generation of yeast strain libraries with combinations of chromosome gains and losses

We first determined how aneuploidy of different regions of the genome combines to impact cellular fitness and drug response by generating two collections of yeast with combinations of chromosome gains or losses. The collections consist of nearly all possible single and double chromosome gains and losses (Fig. 1A). To engineer chromosome gain or loss, chromosome-specific missegregation was first induced via a galactose-inducible centromere-proximal promoter, followed by selection for the specific aneuploidy (Fig. EV1A). We selected for chromosome gains via recombination events that restore the function of an auxotrophic marker. Chromosome loss was generated by selecting against the *URA3* gene using 5-fluoroorotic acid (5-FOA) (Fig. EV1A). The chromosome gain collection includes most of the possible single (14/16) and double disomic (90/136) strains in haploid cells; each was confirmed by qPCR (Appendix Table S1). The chromosome loss collection contains every possible single or double monosomy in diploid cells (Fig. EV1B). Since chromosome loss cannot be continuously selected for, this collection is inducible to generate new aneuploid cells prior to each experiment. These libraries currently constitute the largest collections of engineered complex aneuploid cells.

We first evaluated the libraries by measuring the growth of the engineered aneuploidy strains under optimal growth conditions using a high-throughput assay, where colony size was quantified after 24 h. The growth measurements of single disomies and monosomies highly correlate with those previously observed (Beach et al, 2017) (Fig. EV1C). The previously published measurements were performed in a strain background containing a mutation in the *SSD1* gene, which has been reported to sensitize cells to aneuploidy phenotypes (Hose et al, 2020). However, we observe similar results in a different, wild-type *SSD1* background (S288c), indicating that strain background does not substantially alter the relative growth of aneuploid yeast. The ability to reproduce the previously published correlations between gene number and phenotypic strength and the differences between chromosome gain and loss attest to the quality of these strain collections.

We next determined the impact of combining aneuploid chromosomes on cellular fitness. Growth of complex aneuploidies negatively correlated with the number of open reading frames (ORFs) on the aneuploid chromosomes for both chromosome gain and loss (Fig. 1B). While most of the single and double monosomies are lethal, nearly all of the double disomies are viable, with the sole exception of those involving chromosome 6 (Figs. 1B

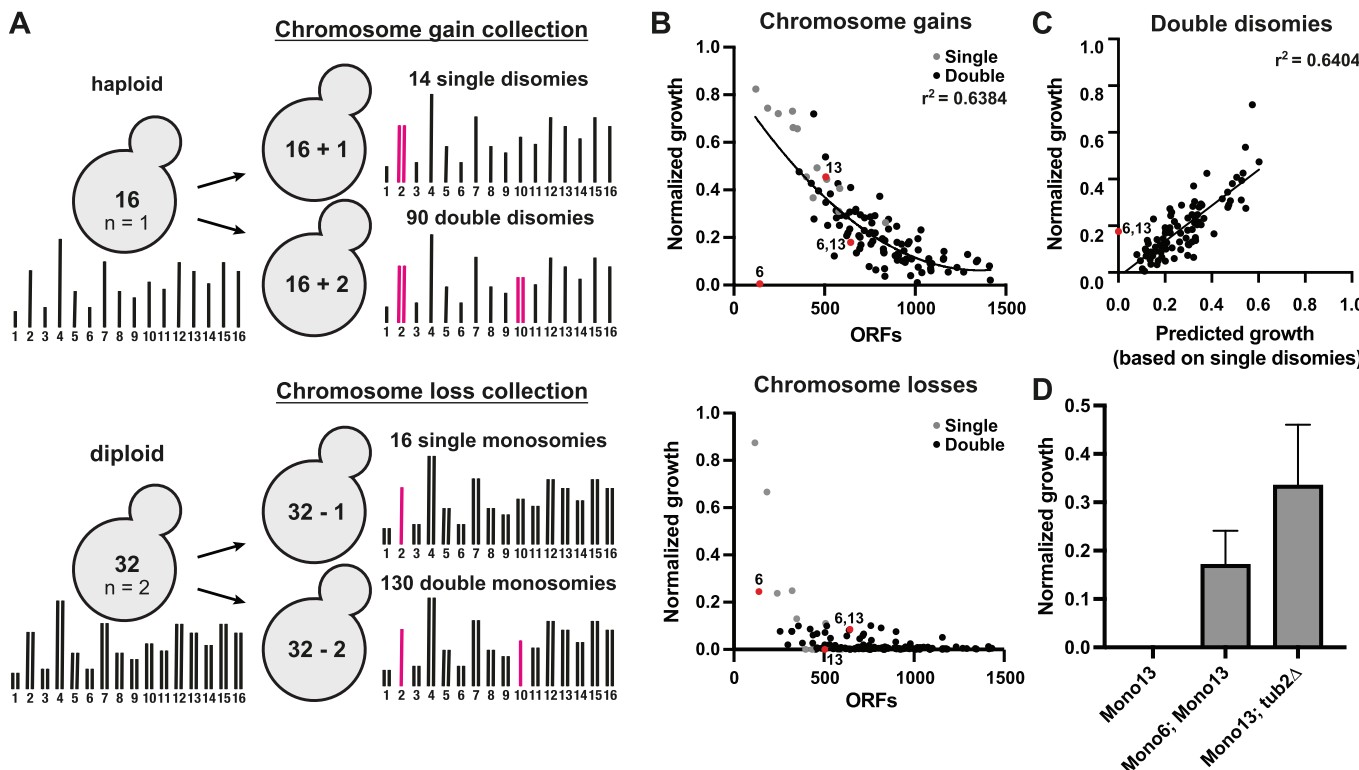

**Figure 1. Strong genetic interactions between aneuploid chromosomes are rare.**

(A) Schematic overview of the chromosome gain and chromosome loss collections. (B) Quantified growth of the chromosome gain and chromosome loss collections. The averages of four independent high-throughput growth assays were normalized to WT. Colony intensities were quantified after 24 h on YPAD plates for chromosome gains, and 72 h on plates containing 5-Fluoroorotic Acid (5-FOA) for chromosome losses. Single disomies/monosomies are depicted in gray, double disomies/monosomies in black. r2 values are from quadratic fits. (C) Correlation between growth of the double disomies and growth predictions based on the single disomies ($P < 1.0 \times 10^{-15}$). Growth was predicted by calculating the product of the single disomy growth. r2 values are from simple linear regression, and P values are from F-tests. (D) Quantification of colony sizes on YPAD plates after 48 h normalized to a euploid diploid strain ($n = 3$, biological). Means and standard deviations are shown. Source data are available online for this figure.

and EV1B). The largest outlier is the combination of chromosomes 6 and 13 (Fig. 1C). Chromosome 6 disomy is lethal on its own due to the presence of an extra copy of the β-tubulin *TUB2* gene (Anders et al, 2009). However, chromosome 6 disomy becomes viable with the additional gain of chromosome 13. This is consistent with previous observations where selection of chromosome 6 gain resulted in a strain that spontaneously gained the additional disomies of chromosomes 1 and 13 (Torres et al, 2007). Chromosome 13 contains the genes for α-tubulin, and overexpression of α-tubulin rescues the lethality of chromosome 6 disomy (Anders et al, 2009). For the chromosome loss collection, one of the most prominent genetic interactions is again between chromosomes 6 and 13 (Fig. 1D). Chromosome 13 monosomy is lethal on its own, but the additional loss of chromosome 6 rescues viability (Fig. 1D). The heterozygous deletion of *TUB2* is sufficient to rescue chromosome 13 monosomy, indicating that excess β-tubulin relative to α-tubulin is the genetic basis behind these chromosome copy number interactions (Fig. 1D).

To identify any additional genetic interaction between chromosome gains, we calculated growth predictions for all chromosome combinations based on the product of the growth of the individual single aneuploidies. In general, the growth of the double disomies correlates well ($r^2 = 0.64$) with the calculated growth predictions

(Fig. 1C). To measure genetic interaction between aneuploid chromosomes, we calculated an epistasis score by subtracting predicted growth from the measured growth of the chromosome combinations. Values near zero reflect additive epistasis and a lack of genetic interaction. Positive and negative scores indicate positive and negative synthetic genetic interactions, respectively. The epistasis scores adhered to a normal distribution (Kolmogorov-Smirnov test for normality), indicating that most combinations do not deviate substantially from the expected growth (Fig. EV1D). Among the 90 chromosome combinations analyzed, we identified only 3 positive (z-score >2) and 2 negative (z-score < −2) chromosome copy number interactions (Fig. EV1D). We conclude that most combinations of aneuploid chromosomes show additive epistasis and that positive or negative genetic interactions (positive or negative epistasis) are rare in the absence of selective pressure.

## The strongest drug-resistance phenotypes come from strains with multiple aneuploid chromosomes

We next used the chromosome gain library to determine if combinatorial interactions between aneuploid chromosomes contribute to improved growth in stress conditions. We measured growth in the presence of four drugs that induce cellular stresses associated with

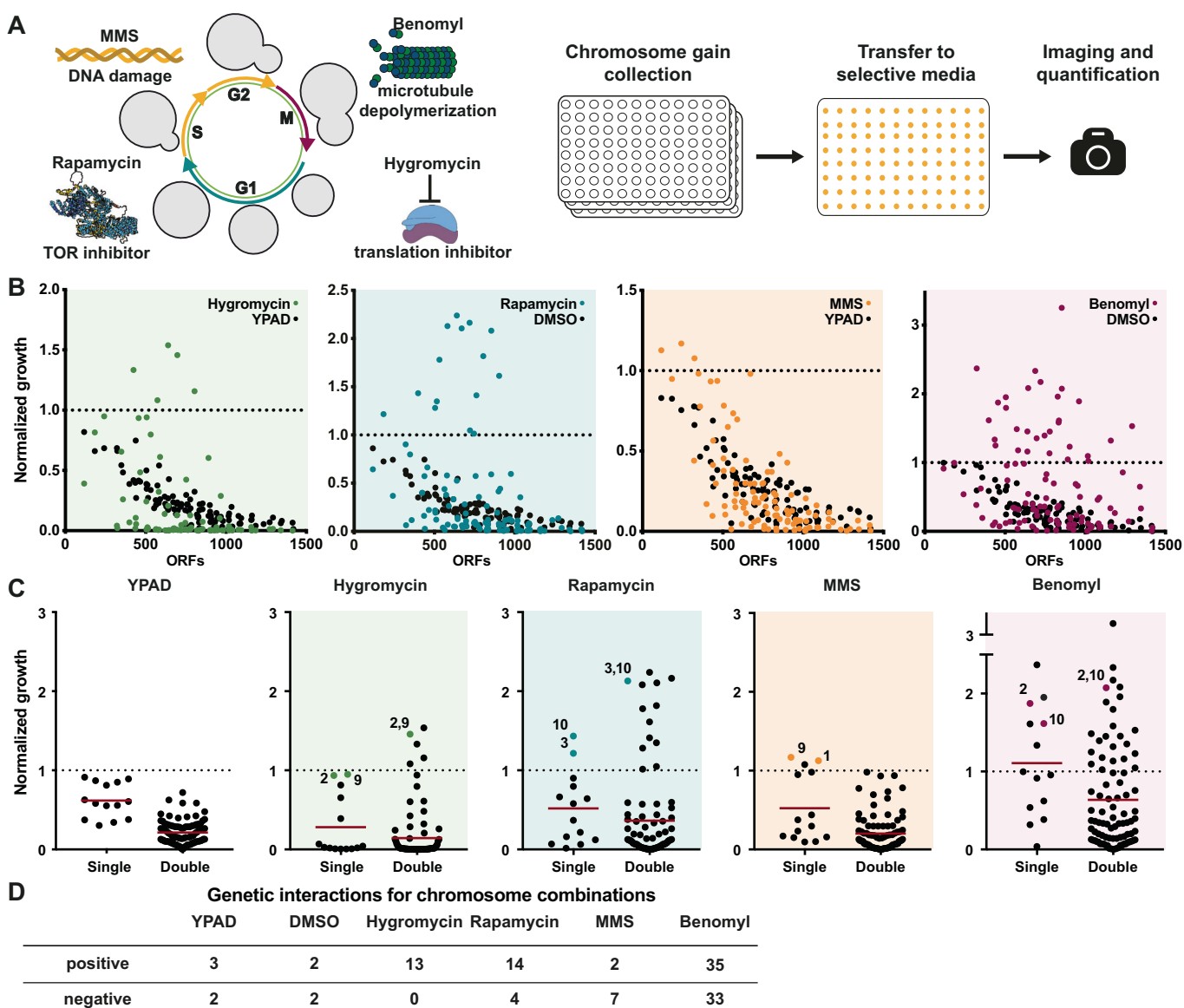

**Figure 2. The strongest drug-resistance phenotypes come from complex aneuploidy.**

(A) Schematic of the workflow for plate-based high-throughput growth assays under selective conditions. Cells were grown in liquid cultures before being transferred to condition-specific agar plates. Images for colony intensities were taken after 24, 36, or 48 h. (B) Averaged growth of two independent high-throughput growth assays. Control plates were quantified after 24 h; hygromycin (40 μg/ml) and rapamycin (5 nM) plates after 48 h; and MMS (0.02%) and benomyl (17.5 μg/ml) plates after 36 h. All quantifications were normalized to a haploid WT. (C) Growth measurements of single disomies (Single) and double disomies (Double) across different treatments. Mean of each group is indicated by a red line. The YPAD control is an average of YPAD and YPAD + DMSO plate growth. The dashed line indicates WT growth. (D) Epistasis scores were calculated by subtracting the predicted growth (the product of the growth of the single disomies) from the measured growth. Genetic interactions are based on the normal distribution of epistasis scores without selective pressure. Positive genetic interactions have an epistasis score greater than the median plus 2 standard deviations and negative interactions have an epistasis score less than the median minus 2 standard deviations. Source data are available online for this figure.

aneuploidy: hygromycin B (inhibitor of ribosomal polypeptide synthesis), rapamycin (TOR pathway inhibitor), methyl methanesulfonate (MMS, DNA damage inducer) and benomyl (microtubule depolymerizer) (Fig. 2A). Although resistance mechanisms for these drugs have been studied extensively, we aimed to identify how drug resistance could be obtained by combining copy number changes of multiple regions in the genome. Drug concentrations were titrated to reduce wild-type growth by ~80%. Growth was unaffected by the solvent used to resuspend the drugs (Fig. EV1E). Aneuploidy

substantially altered the cellular response to hygromycin, rapamycin, and benomyl, as growth no longer scaled with the number of genes on aneuploid chromosomes (Fig. 2B). By contrast, MMS treatment showed fewer aneuploidy-associated phenotypes. The percentage of strains displaying substantial relative resistance (normalized growth in the presence/absence of the drug >2) across the entire library varies greatly between the different drugs, ranging from 59% for benomyl to 9% for MMS (Fig. EV2B,C). We conclude that the capacity for aneuploidy to provide resistance is drug dependent.

In addition to drug resistance, we also observed frequent drug sensitivity. This was most pronounced for hygromycin, where over 78% of strains showed a greater than twofold relative decrease in growth in the presence of the drug (Fig. EV2B). Additionally, we measured sensitivity to two stress conditions other than drugs: 39 °C for heat stress and 18 °C for cold stress. Neither of these conditions greatly influenced cell growth in the euploid strain. However, heat stress led to a twofold decrease in growth for 66% of the disomic strains, which scaled with the degree of aneuploidy (Fig. EV2A–C). These results suggest that aneuploid yeast are especially sensitive to hygromycin and heat stress, which implicates translation inhibition and protein misfolding as potential vulnerabilities in aneuploid cells. This would agree with previous reports that aneuploid strains frequently exhibit decreased ribosome levels and proteotoxic stress (Oromendia et al, 2012; Terhorst et al, 2020).

We next wanted to determine if combinations of aneuploid chromosomes show stronger phenotypes than individual chromosomes. For hygromycin, rapamycin and benomyl, specific chromosome combinations had the highest levels of drug resistance, despite their negative impact on fitness in the absence of the drugs (Fig. 2C). Notably, absolute resistance to hygromycin was observed only for chromosome combinations, as none of the single disomies grew better than the euploid strain in the presence of hygromycin. We measured genetic interactions between chromosomes in the same manner as for the untreated cells by comparing the growth of the chromosome combinations to the predicted growth based on the single disomies (Fig. EV2D). For all four drugs, the epistasis scores no longer fit a normal distribution, suggesting more frequent deviations from the predicted growth (Fig. EV2E). Indeed, after using the same threshold for epistasis scores as we applied to the untreated condition, there were substantially more positive and negative genetic interactions following drug treatment as compared to untreated cells (Fig. 2D). Overall, we conclude that the strongest drug-resistance aneuploidy phenotypes typically come from the combined effects of multiple genes on different chromosomes and that selective pressure increases the frequency of chromosome copy number interactions.

## Benomyl resistance correlates with α-tubulin expression in aneuploid strains

Surprisingly, a majority (59%) of disomic chromosome combinations were resistant to the microtubule depolymerizing factor benomyl (Fig. EV2B,C). This high percentage results from 6 of the 14 tested single disomic chromosomes conferring substantial resistance to benomyl (Figs. 2C and 3A). Although it has previously been reported that many disomies lead to benomyl sensitivity (Sheltzer et al, 2011; Torres et al, 2007), the resistance appears to outweigh the sensitivity for most chromosome combinations. Neither benomyl sensitivity nor resistance is therefore a "general" aneuploidy phenotype but is instead karyotype specific. Aneuploidy frequently leading to resistance to microtubule inhibitors was not specific to benomyl, as four of the same aneuploid chromosomes provided resistance to another microtubule depolymerizing drug, nocodazole (Fig. EV3A). We first focused on chromosome 13, as it contains both α-tubulin genes (TUB1 and TUB3) and α-tubulin overexpression has been shown to result in a mild increase in benomyl resistance (Schatz et al, 1986). Heterozygous deletion of TUB3 in the chromosome 13 disomic strain eliminated the

resistance phenotype, demonstrating the contribution of this gene to the resistance (Fig. EV3B). To determine if α-tubulin protein abundance contributes to benomyl resistance in the other aneuploid strains, we measured the expression levels in all 14 disomic strains via quantitative western blots. We found that multiple single disomies alter α-tubulin expression levels (Fig. 3B). The measured α-tubulin expression levels correlated well with those measured by proteomics of aneuploid yeast strains, demonstrating that these differences occur independently of the measurement method and strain background (Fig. EV3C,D; (Muenzner et al, 2024; Dephoure et al, 2014)). Elevated α-tubulin expression levels highly correlated with the degree of benomyl resistance for the single disomies, suggesting that this is a major mechanism of resistance (Figs. 3C and EV3E). Intriguingly, only chromosome 12 disomy is extremely sensitive to benomyl and is also the only disomy that shows a substantial decrease in α-tubulin expression. The correlation with benomyl resistance was much weaker for β-tubulin expression (Figs. 3C and EV3F,G). To directly measure how α-tubulin abundance affects growth on benomyl, we engineered diploid yeast with different copy numbers of the TUB3 gene, ranging from 0 to 4 copies. Decreasing TUB3 copy number decreases α-tubulin protein levels and causes benomyl sensitivity, whereas additional copies of the gene increase protein abundance and lead to benomyl resistance (Fig. EV3H,I). These results demonstrate that mild changes in tubulin expression levels can greatly alter the response to microtubule inhibitors and that there are multiple genes across the genome that can alter tubulin expression levels when present at an extra copy.

To identify candidate resistance genes on the aneuploidy chromosomes, we started by looking at gene ontology (GO) terms associated with the molecular target (tubulin), associated phenotype (CIN), and drug-sensitivity screens of the yeast deletion collection. None of these terms showed a significant enrichment on the resistant aneuploid chromosomes (Appendix Fig. S1A). Furthermore, there were between 64 and 195 candidate genes per chromosome identified this way (Appendix Fig. S1B). To further refine the list of candidate genes, we focused on genes with substantial increases in RNA and protein levels on the aneuploid chromosomes (Torres et al, 2007; Dephoure et al, 2014). Applying an overexpression threshold of 1.5-fold for candidate proteins only reduced the list by 32%. We therefore sought to identify chromosomal regions responsible for drug resistance to decrease the number of candidate resistance genes.

## Identification of chromosome arms necessary for aneuploidy-based drug resistance

We next aimed to determine which regions of particular chromosomes provide resistance when gained by a single copy. We focused primarily on benomyl resistance, since six individual disomies led to strong resistance to the drug. This provided multiple examples to systematically examine how many regions/genes typically contribute to positive aneuploidy phenotypes for single aneuploid chromosomes. One possibility is that a single strong resistance gene on each chromosome could be sufficient to give a substantial fitness advantage under harsh drug conditions (single-gene model). Another possibility is that multiple mildly beneficial genes on a single chromosome may act together to create strong resistance phenotypes (combinatorial model) (Fig. 4A).

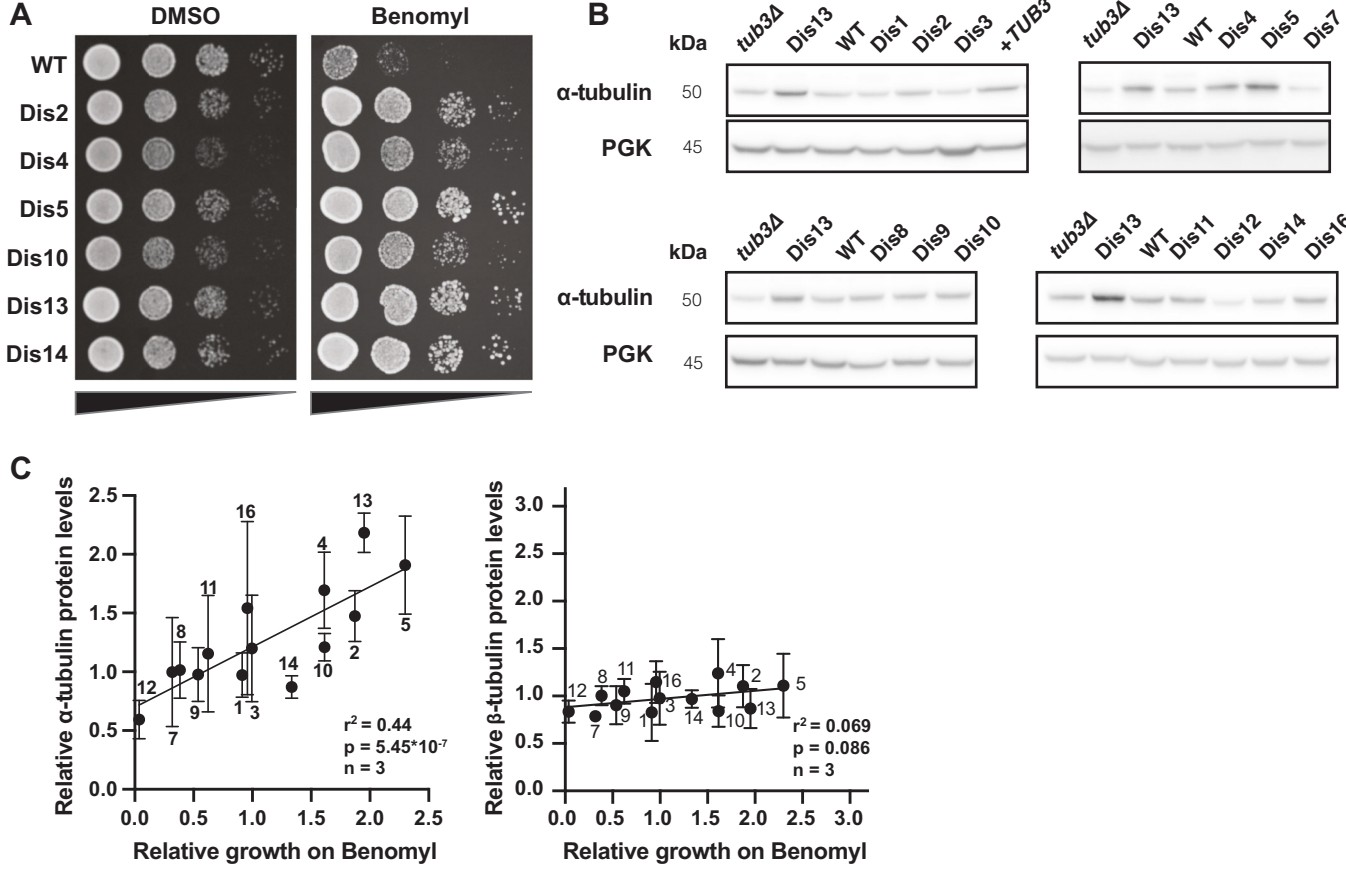

**Figure 3. Benomyl resistance correlates with alpha-tubulin expression in aneuploid strains.**

(A) Tenfold serial dilution analysis of benomyl-resistant single disomies on control (YPAD + DMSO, 24 h) and treated (YPAD + 20 μg/ml benomyl, 48 h) plates. (B) Western blot analysis of α-tubulin expression levels of single disomies. Pgk1 is shown as a loading control. (C) Western blot quantifications of α-tubulin ($n = 3$, biological, $P = 5.45 \times 10^{-7}$) and β-tubulin ($n = 3$, biological, $P = 0.0858$) expression levels for all single disomies. Strains were grown in YPAD and harvested in log-phase. Quantified bands were normalized to the Ponceau-stained membrane and the haploid WT. Relative growth on benomyl denotes the average growth of single disomies from two independent high-throughput colony size assays. $r^2$ values are from simple linear regression, and $P$ values are from F-tests. Means and standard deviations are shown. Source data are available online for this figure.

To determine which regions of chromosomes have negative, positive, or neutral effects on growth with and without benomyl, we implemented a method that deletes part of a chromosome from the end and replaces it with a selection marker and an artificial telomere (Linder et al, 2017) (Fig. 4B). We first generated single arm disomies of all of the benomyl-resistant single disomic chromosomes. The deletion of one of the two copies of the chromosome arms was confirmed by qPCR. As with whole chromosome disomies, arm-level disomies have decreased growth rates proportional to the number of aneuploid ORFs under standard conditions (Fig. 4C). Benomyl resistance was frequently associated with a single arm. For chromosomes 2, 4, 10, and 14, disomy of the larger arm was sufficient to fully recapitulate the benomyl resistance, whereas chromosome 13 only showed resistance with disomy of the shorter arm (Fig. 4D). Both α-tubulin genes (*TUB1* and *TUB3*) are on the short (left) arm of chromosome 13. Intriguingly, neither of the chromosome 5 arm disomies recapitulated the benomyl resistance of the whole chromosome disomy. Chromosome 5 left arm disomy shows no resistance, while the right arm disomy is only mildly resistant.

Therefore, the benomyl resistance for chromosome 5 results from at least one gene on each of the two chromosome arms (Fig. 4D). Overall, the correlation between growth with and without benomyl for the arm-level disomies was not significant, demonstrating that benomyl resistance is not the result of slow growth or a general aneuploidy phenotype (Appendix Fig. S2A). Instead, resistance likely results from the increased expression of one or more specific genes on the aneuploid chromosome arms.

Next, we used our arm-level aneuploidies to analyze resistance or sensitivity to the three other drugs tested: hygromycin, MMS, and rapamycin (Appendix Fig. S2B). Similar to benomyl, resistance to MMS, rapamycin, and hygromycin was primarily associated with single chromosome arms. By contrast, arm-level aneuploidies generally did not fully recapitulate increased drug-sensitivity phenotypes (Fig. 4E; Appendix Fig. S2B,C). Instead, both arms often showed partial sensitivity. This is also the case for the only chromosome that showed high sensitivity to benomyl, chromosome 12 (Fig. 4D). These results suggest that aneuploidy-based drug-sensitivity results from the combined effects of more genes than drug resistance. Surprisingly, in some examples, drug sensitivity

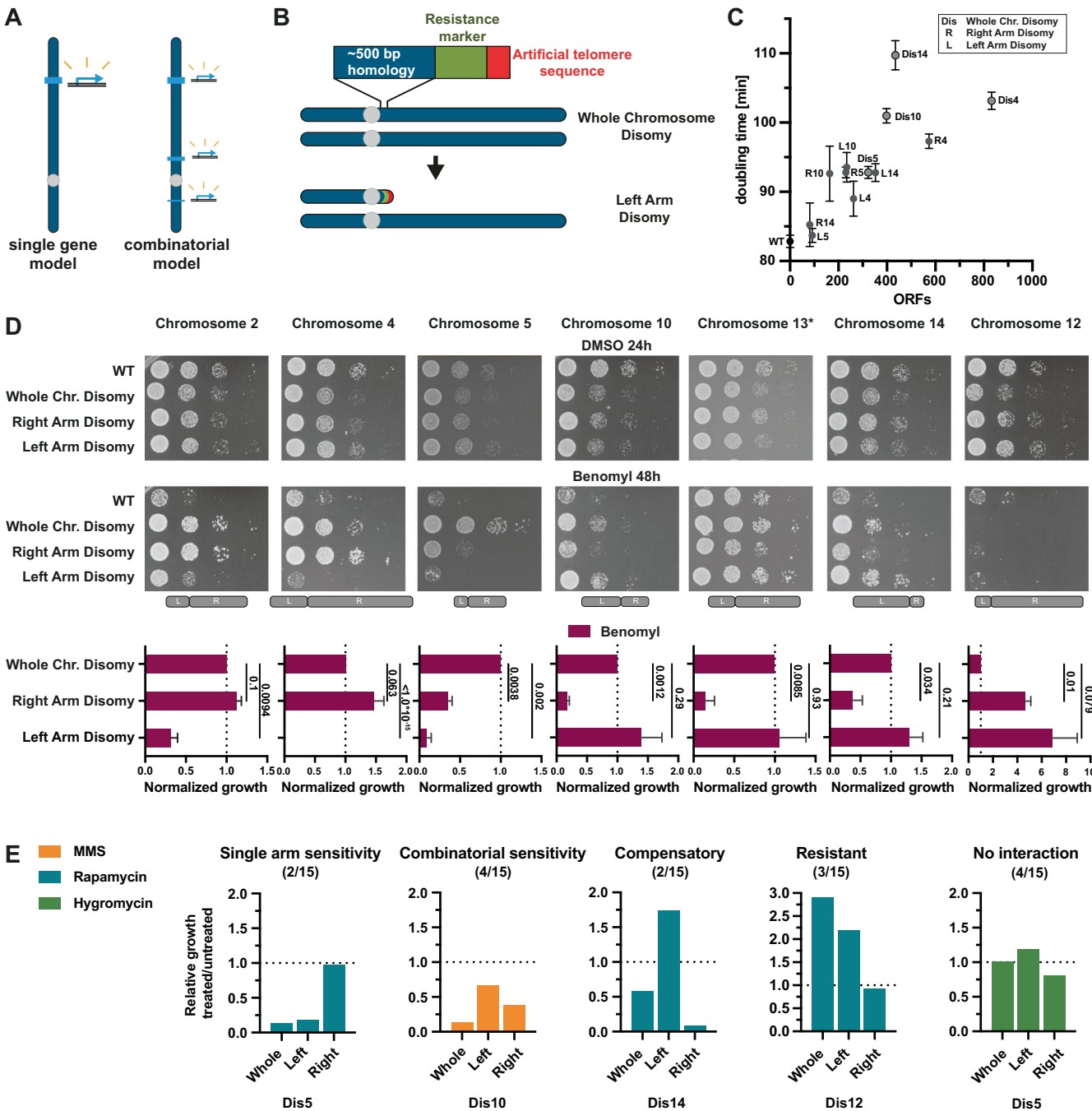

**Figure 4. Identification of chromosome arms necessary for aneuploidy-based drug resistance.**

(A) Schematic of two models for the genetic basis behind aneuploidy-driven drug resistance. (B) Method of chromosome arm deletion. Chromosome arms are deleted by homologous recombination with a 500 bp homology sequence connected to a resistance marker and a telomeric seed sequence. (C) Doubling times of whole chromosome and single arm disomies for chromosomes 4, 5, 10, and 14 (n = 3, technical). ORFs depict the remaining aneuploid ORFs after chromosome arm deletions. Means and standard deviations are shown. (D) Tenfold dilution series and quantification of whole chromosome and single arm disomies (n = 3, biological). Control plates (YPAD + DMSO) were imaged after 24 h, and treated plates (YPAD + 25 µg/ml benomyl) after 48 h. * Indicates a lower benomyl concentration (20 µ/ml benomyl). The second dilution was quantified and normalized to WT, with the exceptions of chromosome 13 (third dilution) and chromosome 12 (first dilution). P values are from one-way ANOVA, uncorrected Fisher's LSD test. Means and standard deviations are shown. (E) Quantification of tenfold serial dilutions treated with MMS (0.04%), rapamycin (10 nM), or hygromycin (60 µg/ml). Relative growth was calculated as the ratio of normalized growth on treated plates (48 h) to normalized growth on control plates (24 h). Representative examples for single arm sensitivity, combinatorial sensitivity, compensatory sensitivity, resistance, and no interactions are shown. Images and quantifications of all tenfold serial dilutions are depicted in Appendix Fig. S2B, C. Source data are available online for this figure.

from one chromosome arm was compensated for by the resistance from the second chromosome arm (Fig. 4E; Appendix Fig. S2B,C).

## Most aneuploidy-based drug-resistance phenotypes come from the combined effect of multiple genes

Since most of the resistance phenotypes were associated with the larger arms, we wanted to determine how many regions/genes on these arms contribute to benomyl resistance. We therefore made multiple shorter deletions to the benomyl-resistant disomies to identify regions of drug resistance. For example, seven additional cuts were made on the right arm of chromosome 4 to identify regions of interest. As the size of the chromosome decreased, growth under standard conditions gradually increased (Fig. 5A). This further demonstrates that the negative effects on fitness for aneuploid strains result from many genes across the length of a chromosome (Bonney et al, 2015). Conversely, benomyl resistance decreased in a stepwise manner following the deletion of three independent regions on the right arm (Fig. 5A). For five of the six resistant chromosomes, we identified at least two regions that contributed to resistance, with the most being 5 regions for chromosome 5 (Figs. 5C and EV4A). Interestingly, three of the benomyl-resistant chromosomes also had regions that increased drug sensitivity (Figs. 5A,C and EV4A). The strength of drug resistance is therefore a balance between regions that confer resistance and sensitivity.

To determine if the regions that lead to benomyl resistance also increase α-tubulin protein levels, we examined the segmental deletions of chromosome 5 in the disomic strain. Despite both arms of chromosome 5 contributing to benomyl resistance, only the right arm was necessary for the increased α-tubulin levels. Removal of a single region on the right arm decreased the α-tubulin levels back to euploid levels (Fig. EV4B). The identified 60 kb region also contributes to benomyl resistance. This region does not contain any genes with an associated tubulin/microtubule gene ontology term. None of the other 4 resistance regions contributing to benomyl resistance impacted α-tubulin protein levels, indicating that these regions confer benomyl resistance through alternative mechanisms.

We next aimed to identify the resistance genes within the identified regions. The regions each contain between ~30 and 100 genes. We identified candidate genes by looking for: sensitivity to benomyl in high-throughput screens of the yeast knockout collection, essential genes, and genes that have GO Terms associated with CIN and tubulin/microtubules (Appendix Fig. S1B). Candidate genes were tested via heterozygous deletions in the disomic strains (Appendix Fig. S3A). For the two regions on chromosome 4 associated with the strongest benomyl resistance, we identified single gene deletions that decreased resistance to a similar extent as deleting the entire chromosomal region (Fig. 5B,C). In total, we identified 9 genes that decrease the resistance of aneuploid chromosomes when deleted heterozygously (Appendix Figs. S3A and S4). In addition to *TUB3* on chromosome 13, we identified two additional genes associated with tubulin (*GIM4*-chr.5, *PFD1*-chr.10). Both of these genes are members of the prefoldin complex, which is a chaperone that promotes the proper folding and assembly of α- and β-tubulin (Vainberg et al, 1998). One resistance gene is involved in microtubule stabilization (*BIM1*-chr.5). Interestingly, two of the resistance genes have recently been implicated in the spindle position checkpoint (*SWR1*-chr.4, *MCK1*-chr.14) (Rathi et al,

2022; Caydasi et al, 2023). Another resistance gene, *TOF1*-chr.14, is primarily associated with the protection of the DNA replication fork. However, it shows a synthetic lethal interaction with *BIM1*, indicating an additional role in chromosome segregation (Tong et al, 2004; Pan et al, 2004). Overall, we have identified many genes across the genome with a variety of different functions that contribute to aneuploidy-based benomyl resistance.

Previous studies have only identified two or three genes responsible for resistance phenotypes for individual perturbations, prohibiting any significant analysis of the characteristics of genes that can contribute to aneuploidy phenotypes either in yeast (rev. in Tsai and Nelliat, 2019) or human cells (Adell et al, 2023; Ippolito et al, 2021). We assessed whether the 9 genes identified shared common features using GO enrichment, RNA expression levels, and protein abundance. Intriguingly, GO overrepresentation analysis of the nine identified benomyl resistance genes did not reveal any significant enrichment terms as measured by adjusted *P* values (Appendix Fig. S3D). In addition, the highest-ranked hit, "microtubule cytoskeleton organization", only contained three out of nine genes. This highlights the diversity of gene functions that can combine to contribute to aneuploidy-based drug resistance.

Different genes can have substantial differences in the degree to which their expression increases when an additional copy is gained on an aneuploid chromosome (Stingele et al, 2012; Dephoure et al, 2014; Muenzner et al, 2024; Hwang et al, 2021; Sousa et al, 2019). We therefore hypothesized that genes that contribute to beneficial aneuploidy phenotypes would be those that have their expression levels more substantially altered by aneuploidy. Analysis of the increase in RNA expression levels for genes on aneuploid chromosomes showed a small, yet significant increase for the 9 benomyl resistance genes when compared to the rest of the genes on these chromosomes (Torres et al, 2007; Appendix Fig. S3E) However, this may reflect a bias in the selection of candidate genes, as there was no significant difference between the tested genes that either did or did not contribute to benomyl resistance (Appendix Fig. S3F). For the seven benomyl resistance genes where increased protein levels on aneuploid chromosomes have been measured, there were no significant differences in expression levels (Appendix Fig. S3E,F). We conclude that genes affecting aneuploidy phenotypes are not enriched for genes with more substantial increases in expression on aneuploid chromosomes.

In addition to the disomic chromosomes that lead to benomyl resistance, we also identified chromosomal regions that lead to rapamycin resistance on chromosome 10. Strains with disomy of chromosome 10 have previously been shown to be rapamycin resistant (Torres et al, 2007). This was speculated to be due to the presence of the *TOR1* gene. Heterozygous deletion of *TOR1* in a chromosome 10 disomic strain only partially decreased rapamycin resistance, indicating that there are more genes on chromosome 10 that contribute to the phenotype (Appendix Fig. S3B). Furthermore, both chromosome arms contribute to rapamycin resistance (Appendix Fig. S3C). We conclude that the combined overexpression of multiple genes leading to strong phenotypes is a common feature of aneuploidy.

## The strongest aneuploidy phenotypes result from synergistic effects

Chromosome 5 disomy has the highest level of benomyl resistance and is the only chromosome where both arms contribute to the

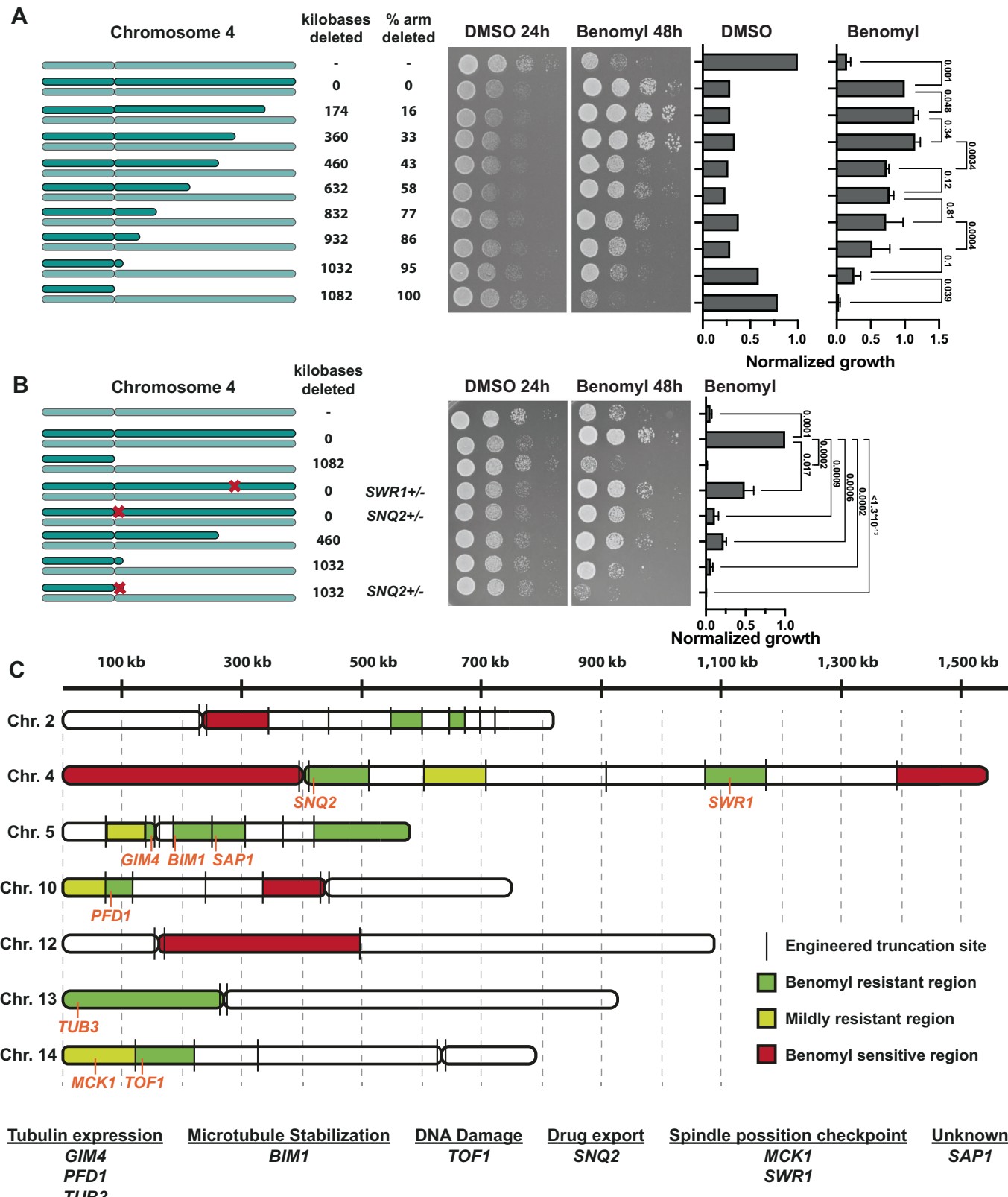

◄  **Figure 5. Most aneuploidy-based drug-resistance phenotypes come from the combined effects of multiple genes.**

(A) Representative image of 10-fold serial dilution and quantification of partial chromosome arm deletions of the right arm of chromosome 4. For control plates (YPAD + DMSO) the 2nd spot was quantified after 24 h and for treated plates (YPAD + 25 μg/ml benomyl) the 2nd and 3rd spots were quantified and averaged after 48 h ($n = 3$, biological). P values are from one-way ANOVA, uncorrected Fisher's LSD test. Growth on DMSO was normalized to a haploid WT, and growth on benomyl to a strain disomic for chromosome 4. Means and standard deviations are shown. (B) Tenfold serial dilution and quantification of partial chromosome arm deletions and heterozygous deletions of genes contributing to benomyl resistance. For treated plates (YPAD + 25 μg/l benomyl) the 3rd spot was quantified after 48 h and normalized to the strain disomic for chromosome 4 ($n = 3$, biological). P values are from one-way ANOVA, uncorrected Fisher's LSD test. Means and standard deviations are shown. (C) Schematic depiction of benomyl-resistant and sensitive chromosome regions. Identified resistance genes are indicated in orange at their approximate chromosomal location. Source data are available online for this figure.

phenotype (Fig. 4D). In addition, the degree of resistance for the whole chromosome was substantially higher than either of the two individual arm-level disomies, suggesting a synergistic interaction between the two arms. To look at this more systematically, we predicted the strength of resistance for whole chromosome aneuploidy based on the resistance of the individual arm-level aneuploidies (Fig. 6A). Although the other single disomies displayed resistance in line with the predictions, benomyl resistance for chromosome 5 was substantially higher than predicted. This demonstrates a synergistic interaction between the two arms.

We next wanted to determine if the strongest phenotypes for double disomic strains also result from synergistic interactions. We focused on chromosome combinations where the single disomies on their own were benomyl resistant. We initially observed that the combined disomies of chromosomes 2 & 4 and 2 & 13 are exceptionally strong, showing robust resistance even at higher benomyl concentrations (Fig. 6B). These combinations are as resistant as a deletion mutation known to be highly resistant to benomyl (Schibler et al, 2016), despite substantial growth defects for aneuploid strains in the absence of the drug (Fig. 6C). Chromosome 13 disomy shows little-to-no growth at this concentration of benomyl yet substantially increases the resistance when combined with chromosome 2 disomy, clearly demonstrating a positive genetic interaction. These high levels of resistance do not result from further increases in α-tubulin expression levels (Fig. EV5A). Looking at combinations with chromosome 2 specifically, we observed a wide range of resistance phenotypes, ranging from highly positive to highly negative interactions (Figs. 6B and EV5B). Strikingly, a negative genetic interaction between chromosomes 2 and 14 results in the near-complete loss of the benomyl resistance (Fig. 6B,D). Almost all combinations with chromosome 14 grow worse than the single disomies (Fig. EV5C), indicating a general negative effect of chromosome 14 combinations and benomyl resistance. We also observed a synergistic interaction between chromosomes 3 and 12 for rapamycin resistance, demonstrating that such genetic interactions are not benomyl-specific (Fig. EV5D). We conclude that most aneuploidies display additive effects between genes, regions, and chromosomes. However, the strongest beneficial growth phenotypes often come from synergistic interactions.

Next, we tested if double disomies have a continuous advantage over single disomies during ongoing selective pressure. We co-cultured chromosome 2 & 4 double disomic strains labeled with mNeonGreen with either euploid, chromosome 2 disomy or chromosome 4 disomy strains labeled with mCherry and determined their relative abundance daily (Fig. EV5E). Under non-selective conditions, the double disomic strain was rapidly outcompeted by all of the other strains. By contrast, in the presence of benomyl, the double disomic strain initially outcompeted each of the other strains. Interestingly, the chromosome 2 disomy strain was never fully outcompeted and stabilized at an 80/20 (dis2&4/dis2) ratio. qPCR analysis of individual colonies from the day 3 populations showed that the chromosome 2 disomy strains gained an additional copy of chromosome 13 (5/5 colonies) (Fig. EV5F). This result agrees with our observation that disomies 2 & 4 and 2 & 13 show very similar levels of benomyl resistance due to synergistic chromosome combinations. These results highlight the competitive advantage of synergistic combinations of beneficial aneuploid chromosomes.

## Discussion

In this study, we generated a library of engineered single and double aneuploid yeast strains to systematically determine the genetic drivers of aneuploidy-based drug resistance. In addition, we used a comprehensive approach to identify regions/genes on aneuploid chromosomes that contribute to aneuploidy phenotypes. This approach has numerous advantages, allowing for the identification of: (i) multiple regions on a chromosome that contribute to a phenotype, (ii) both resistant and sensitive regions on a single chromosome, (iii) regions that act synergistically, where each region on its own would not have a strong phenotype. Together, these strategies provide a means to comprehensively identify the genetic determinants of positive and negative selection for aneuploid chromosomes either alone or in combination.

When measuring the resistance of the chromosome gain collection to four different inhibitors, the strongest resistance phenotypes were consistently identified for chromosome combinations. This demonstrates that altered copy numbers of multiple genes combine to create stronger beneficial aneuploidy phenotypes. These additional benefits are observed despite the often substantial negative growth phenotypes for the double aneuploid strains under optimal growth conditions. Therefore, the benefits of combining multiple aneuploid chromosomes generally outweigh the downsides under selective conditions. Furthermore, we saw these positive combinatorial effects for 3 of the 4 drug conditions, suggesting that it is a general property of aneuploidy. This combinatorial behavior of aneuploid chromosomes could explain the frequent occurrence of complex aneuploidy in both adapted yeast populations and cancer karyotypes (Mitelman Database of Chromosome Aberrations and Gene Fusions in Cancer, 2025. https://mitelmandatabase. isb-cgc.org (Rancati et al, 2008; Ravichandran et al, 2018; Peter et al, 2018; Klockner and Campbell, 2024).

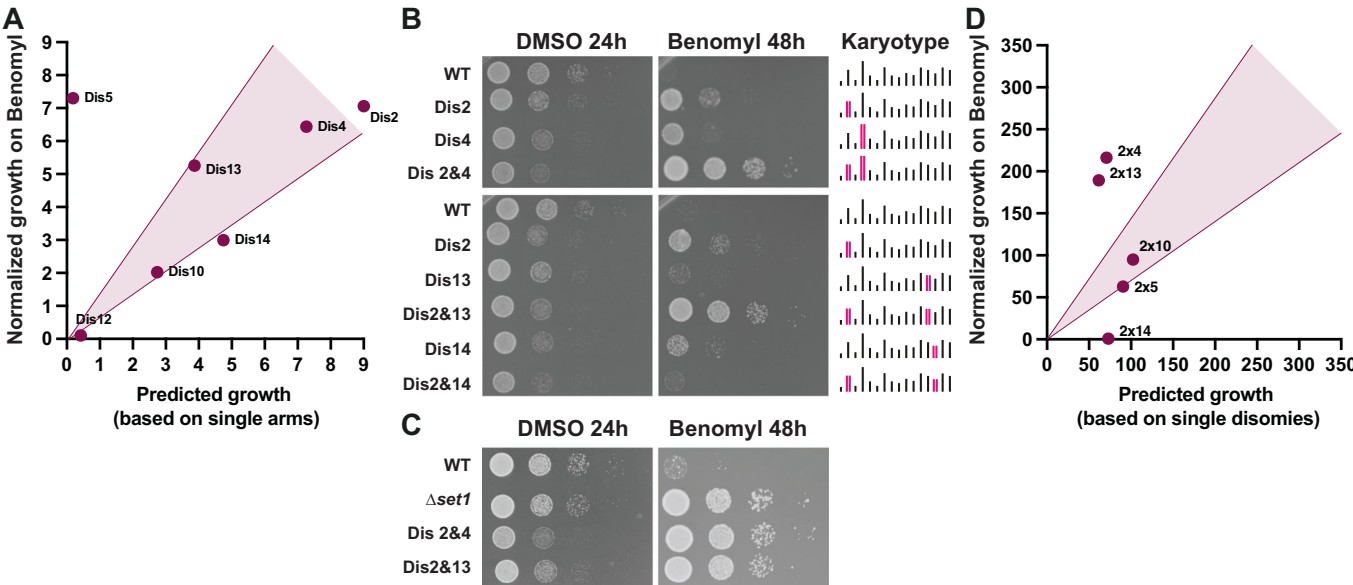

**Figure 6. Synergistic effects contribute to the strongest aneuploidy phenotypes.**

(A) Single disomy growth on benomyl compared to growth predictions based on single arm disomies. Growth was quantified from 10-fold serial dilutions in Fig. 4D. The area highlighted in magenta indicates additive genetic interactions (+/−30% of predicted growth). Disomies that fall below this threshold are considered a negative synergistic interaction, and those above are considered a positive synergistic interaction. (B) Tenfold serial dilution of single and double disomies. Control plates (YPAD + DMSO) were imaged after 24 h, and treated plates (YPAD + 30 μg/ml benomyl) after 48 h. (C) Tenfold serial dilutions of benomyl-resistant set1Δ haploid and two synergistically interacting double disomies (2 & 4, 2 & 13). Benomyl plates contained 30 μg/ml benomyl. (D) Quantification of the tenfold serial dilutions of the double disomies on plates with 30 μg/ml benomyl compared to predicted growth based on the additive resistance of the single disomies (for images, see Fig. EV5B). Combinations in the highlighted area (magenta) are additive genetic interactions (+/−30% of predicted growth). Disomy combinations that fall below this threshold are considered a negative synergistic interaction and those above are considered a positive synergistic interaction. Source data are available online for this figure.

In addition to identifying strong phenotypes for combinations of aneuploid chromosomes, we discovered multiple regions within individual chromosomes that contribute to drug-resistance phenotypes. We identified up to 5 regions on a single chromosome that contribute to drug resistance, emphasizing that multiple genes are typically responsible for aneuploidy phenotypes. This combinatorial action of many genes suggests an advantage for the selection of aneuploidy over other types of genomic alterations. Changing the expression of genes with distinct and complementary functions may provide proliferative benefits that could not be accomplished through single mutations. These data provide substantial support for a "combinatorial genes" explanation for the specific benefits of aneuploidy over other genomic alterations.

In the absence of selection, we did not observe substantial genetic interactions between chromosome gains, with the exception of the combination of chromosomes 6 and 13. However, it is worth noting that even relatively weak interactions have the potential to affect aneuploidy patterns (Ravichandran et al, 2018). By contrast, under selection, the number of genetic interactions between chromosomes was substantially increased for all 4 of the drugs tested. Furthermore, increasing the degree of selection via higher concentrations of benomyl revealed highly synergistic positive interactions between aneuploid chromosomes. We also observed a clear synergistic interaction between the arms of chromosome 5. These results suggest another possible advantage for adaptation through aneuploidy, wherein the altered expression levels of two complementary genes would not be selected for individually due to a lack of phenotype. However, two simultaneous changes to the

genome through a single chromosome missegregation event would enable their combined selection, leading to a strong beneficial phenotype. These results, along with the recently discovered synergistic interaction between chromosomes 3 and 6 in multiazole tolerance of the human pathogen *Candida albicans* (Zhou et al, 2024), demonstrate that positive synergistic interactions are more heavily selected for under higher stress conditions. We also find that combining beneficial aneuploidies can occasionally lead to weaker than expected phenotypes, highlighting that negative genetic interactions also likely impact complex karyotype formation. In cancers, positive and negative genetic interactions between co-occurring aneuploidies have been suggested based on correlations between chromosome arm frequencies in aneuploidy patterns (Ozery-Flato et al, 2011; Ravichandran et al, 2018). Such co-occurrence patterns can be highly predictive of disease progression and chemotherapy resistance in cancer (Shukla et al, 2020). Our results suggest that these patterns are more likely to result from specific selective pressures, rather than more general synergies or incompatibilities between the gain or loss of specific chromosome arms.

The main drivers of beneficial aneuploidy phenotypes are alterations in expression levels resulting from gene copy number changes of specific resistance-driving genes (Selmecki et al, 2009; Chen et al, 2012; Ryu et al, 2016; Linder et al, 2017; Beaupere et al, 2018; Ravichandran et al, 2018; Barney et al, 2021; Pavani et al, 2021; Adell et al, 2023; Zhou et al, 2024; Watson et al, 2024). We observed a significant correlation between growth on benomyl and altered α-tubulin expression levels. In addition to *TUB3* on

chromosome 13, two members of the prefoldin complex (*GIM4*-chr. 5, *PFD1*-chr. 10) contribute to aneuploidy-based benomyl resistance. Intriguingly, chromosome 12 disomy decreases tubulin expression levels and is very sensitive to microtubule destabilization via benomyl. This decrease in tubulin levels for chromosome 12 disomy could also help explain how it rescues microtubule hyperstabilization (Macaluso et al, 2025). We conclude that tubulin expression levels are not tightly regulated in yeast. Furthermore, even relatively small changes in α-tubulin expression levels can substantially influence resistance and sensitivity to microtubule inhibitors. Moreover, the genes contributing to benomyl resistance were not always among the highly expressed genes in both transcriptomics and proteomics datasets, demonstrating that even mild overexpression can lead to substantial phenotypes. These findings underscore the importance of recognizing subtle ( < 2-fold) expression changes, as they have the capacity for causing strong phenotypes.

Although increased α-tubulin expression appears to be a common mechanism of aneuploidy-based benomyl resistance, we also find evidence for additional mechanisms. First, not all of the genes that contributed to benomyl resistance have known functions related to tubulin expression (Appendix Fig. S3D). We identified two components of the spindle position checkpoint (*MCK1*-chr. 14, *SWR1*-chr. 4), implicating this checkpoint in mitigating the effects of microtubule-targeting drugs. We also identified a drug exporter (*SNQ2*-chr. 4) and a microtubule stabilizer (*BIM1*-chr. 5). Importantly, the genes that we identified for resistant chromosome regions were not always the most obvious candidates (Appendix Fig. S3A). *TOF1*-chr. 14 and *SAP1*-chr. 5 have not previously been associated with microtubule functions, but both contribute to benomyl resistance on their respective aneuploid chromosomes. Second, not all of the resistant disomic strains had increased α-tubulin expression (e.g., Dis14, Fig. 3C). Third, not every region of a disomic chromosome that contributes to benomyl resistance increases α-tubulin expression (Fig. EV4B). Finally, we find that disomy combinations with exceptionally high benomyl resistance do not increase α-tubulin levels further than the single disomies (Fig. EV5A). We conclude that different regions on individual chromosomes can combine for beneficial aneuploidy phenotypes through additive or synergistic effects from combining multiple resistance pathways. Additionally, genes that we would have expected to contribute to benomyl resistance based on previous knockout screens often did not have an effect when heterozygously deleted from the disomic strains. The most prominent example is a drug efflux pump located on chromosome 2, Flr1, which did not contribute to benomyl resistance for that chromosome. Deletion of *FLR1* leads to a high degree of benomyl sensitivity (Brôco et al, 1999), yet an extra copy from the disomic chromosome did not affect resistance (Appendix Fig. S3A). Together, these results demonstrate that gene functions determined primarily via loss-of-function mutants may often not correspond with the phenotypes that emerge from increased gene expression due to copy number gain.

While mild overexpression of specific genes can confer drug resistance, the overexpression of many genes across an entire chromosome greatly increases aneuploidy-induced cellular stress (Torres et al, 2007; Sheltzer et al, 2011; Oromendia et al, 2012; Thorburn et al, 2013; Beach et al, 2017; Li and Zhu, 2022). This stress presents potential targets to selectively eliminate aneuploid cells (Cohen-Sharir et al, 2021; Ippolito et al, 2024). However, the potential to leverage aneuploidy-based sensitivities requires a deeper understanding of the underlying mechanisms driving these vulnerabilities. 78% of the strains in the chromosome gain collection were sensitive to hygromycin B, and 66% were sensitive to heat stress (Fig. EV2C). These results provide additional support for decreased ribosome function and proteotoxic stress as hallmarks of aneuploidy (Oromendia et al, 2012; Terhorst et al, 2020). By contrast, inducing DNA damage via MMS did not result in large-scale sensitivity, suggesting aneuploidy-induced genome instability may be more chromosome-specific. Despite the general sensitivity to hygromycin, we identified a few examples of aneuploidy-driven hygromycin resistance. In addition, the sensitivity of one chromosome arm can be masked by resistance of the other arm (Fig. 5C; Appendix Fig. S2B,C). Together, these results demonstrate that regions of the genome that contribute to sensitivity or resistance can fully or partially counteract each other. Sensitivity and negative selection have been predicted to be drivers of cancer karyotype evolution, potentially even more so than positive selection (Jubran et al, 2024).

Given the high prevalence of aneuploidy in cancer, it is an attractive target for identifying exploitable vulnerabilities. This requires a better understanding of the positive and negative forces underlying aneuploid chromosome selection. This work highlights additional challenges for identifying the genes responsible for aneuploidy phenotypes, as they are likely to result from the combined effects of multiple genes.

# Methods

**Reagents and tools table**

| Reagent/resource | Reference or source | Identifier or catalog number |
|---|---|---|
| **Experimental models** | | |
| S288c (BY) (*S. cerevisiae*) | Brachmann CB et al, 1998 | N/A |
| **Recombinant DNA** | | |
| pCC237 | Bähler et al, 1998 | |
| pCC239 | Janke et al, 2004 | |
| pKA52 | Anders et al, 2009 | |
| pGALCEN- JC3-13 | Anders et al, 2009 | |
| pCC658 | Ravichandran et al, 2018 | |
| pCC644 | Ravichandran et al, 2018 | |
| pBS35 | Hailey et al, 2002 | |
| Additional plasmids and more information | This study | Appendix Table S3 |
| **Antibodies** | | |
| (HRP)-conjugated antimouse IgG secondary antibody | Cell Signaling Technology | Cat # 7076 |
| mouse anti-Pgk1 | Thermo Fisher Scientific | Cat # 22C5D8 |
| mouse anti-α-Tubulin | Sigma-Aldrich | Cat # T6074 |

| Reagent/resource | Reference or source | Identifier or catalog number |
|---|---|---|
| mouse anti-β-Tubulin | Developmental Studies Hybridoma Bank | Cat # E7 |
| **Oligonucleotides and other sequence-based reagents** | | |
| PCR primers | This study | Appendix Table S4 |
| **Chemicals, enzymes, and other reagents** | | |
| 5-FOA | Chempur, Karlsruhe, Germany | Cat # 220141-70-8 |
| Benomyl | Sigma-Aldrich | Cat # 45339 |
| Hygromycin B | Carl Roth | Cat # CP12.1 |
| MMS | Sigma-Aldrich | Cat # 129925 |
| Nocodazole | VWR | Cat # ACRO358240100 |
| Rapamycin | Santa Cruz Biotechnology | Cat # sc-3504A |
| Phusion polymerase | New England BioLabs | Cat # M0530L |
| Luna qPCR Mastermix | New England BioLabs | Cat # M3003E |
| Ponceau S | Sigma-Aldrich | Cat # P3504 |
| BSA | Sigma-Aldrich | Cat # A2153 |
| skim milk | Gerbu Biotechnik GmbH | Cat # 68514-61-4 |
| Amersham ECL Prime detection reagent | Cytiva | Cat # RPN2236 |
| RNase A | Sigma-Aldrich | Cat # R6513 |
| Proteinase K | Sigma-Aldrich | Cat # P2308 |
| SYTOX green | Thermo Fisher | 1 Cat # 0768273 |
| Paraformaldehyde | Sigma-Aldrich | Cat # 158127 |
| **Software** | | |
| ImageJ | National Institute of Health | |
| GraphPad Prism 10.0c | Dotmatics | |
| BD FACSDiva 8.0.1 software | BD Biosciences | |
| GOrilla gene ontology tool | http://cbl-gorilla.cs. technion.ac.il | |
| **Other** | | |
| ROTOR HDA robot | Singer Instruments | |
| Synergy S1 plate reader | BioTek Instruments | |
| Benchtop mini-centrifuge | Thermo Fisher Scientific | Cat # 75004061 |
| ChemiDocMP | Bio-Rad | |
| LSRFortessaTM Cell Analyzer | BD Biosciences | |

## Yeast strains and media

All yeast strains and plasmids used in this study are listed in Appendix Tables S2 and S3. All yeast strains are derivatives of BY4741 and BY4742. All yeast strains were cultured in complete synthetic media (CMS) lacking histidine, uracil, lysine, and leucine and cultured at 30 °C, unless otherwise stated. For 5-FOA (Chempur, Karlsruhe, Germany, 220141- 70-8) selection, a

concentration of 1 mg/mL was used. Growth assays were always conducted in yeast extract/peptone containing 40 μg/ml adenine-HCl (YPA) and 2% glucose (YPAD) in liquid cultures and plate-based assays. For high-throughput drug-resistance screens, 17.5 μg/ml benomyl (Sigma-Aldrich), 40 μg/ml hygromycin B (Carl Roth), 0.02% MMS (Sigma-Aldrich), or 5 nM rapamycin (Santa Cruz Biotechnology) were added to YPAD agar plates. YPAD + DMSO agar plates were used as controls for drugs dissolved in DMSO. Genetic manipulations such as insertions and gene deletions were conducted as previously described (Longtine et al, 1998). Strains inducible for aneuploidy were generated using a conditional centromere, as described in Anders et al (2009). Single disomies and double disomies were engineered as described in (Ravichandran et al, 2018). Each disomy was generated independently with two different pairs of auxotrophic markers (URA3 and HIS3; LYS2 and LEU2). For engineering monosomies, one copy of the endogenous centromere was replaced by $P_{GAL1}$-CEN3 URA3. Flow cytometry of selected strains was used to detect potential changes in ploidy and exclude strains that spontaneously diploidized.

## PCR-mediated partial chromosome deletions

PCR-mediated partial chromosome deletions were implemented as described in Linder et al (2017). To generate a deletion, a 500–800 bp fragment homologous to the intended cut-site was combined with a resistance marker (kanMX or hphNT1) and a telomeric seed sequence (Kaboli et al, 2016; Sugiyama et al, 2005; Sugiyama et al, 2008). The telomeric seed sequence (6×5'CCCCAA3') was added at the 3' end of the primer binding the resistance cassette. The homology sequence was PCR amplified from genomic DNA with an added 30 bp homology sequence to the resistance cassette. Fragments were combined using overlap extension PCR.

## High-throughput growth measurements

For monosomies, chromosome missegregation was induced immediately before plating. Strains were grown for 18 h in 2 ml deep-well plates in 200 μl YPAD at 30 °C (shaking). In total, 4 μl were transferred to 150 μl of YPA plus 1% raffinose (YPAR) and grown to log phase (~4 h). 50 μl of YPA medium containing 4% galactose were added and grown for 3 h to induce chromosome missegregation. Samples were plated on 5-FOA agar plates using a ROTOR HDA robot for high-throughput plating (Singer Instruments, Roadwater, UK). Images were taken after 24, 48, and 72 h. The chromosome gain collection was grown in 2 ml deep-well plates in 200 μl CSM lacking histidine, uracil, leucine, and lysine at 30 °C. After 18 h, cultures were diluted 1:10 in 1× PBS and plated with the ROTOR HDA onto condition-specific YPAD plates. Strains were grown at 30 °C (if not indicated otherwise) and images were taken at different time points from 24 to 72 h. Timepoints were selected for quantification based on the highest resolution of growth differences between strains.

## Doubling time measurements

Overnight cultures were grown in CSM lacking histidine, uracil, leucine, and lysine, diluted in YPAD, and grown for 4 h to mid-log phase. Cultures were diluted to an optical density at 600 nm of 0.02

in 200 μl YPAD in 96-well plates. OD measurements were taken every 15 min for 24 h using a Synergy S1 plate reader (BioTek Instruments, Inc., Luzern, Switzerland).

## Serial dilutions

Strains were grown overnight in CSM lacking histidine, uracil, leucine, and lysine. Cultures were diluted in fresh media (OD ~ 0.05) and grown to log phase (OD ~ 0.45), normalized by OD, diluted in 1× PBS (1:10), and plated on condition-specific agar plates.

## High-throughput growth assay quantification and analysis

Images of yeast colonies on plates were quantified using ImageJ (National Institute of Health). Integrated densities of each spot area and the surrounding area were quantified. Background subtraction was done for each spot area individually by subtracting the surrounding intensity from the colony intensity (Dataset EV1). For comparing the growth of chromosome loss strains to chromosome gain strains, 10 representative chromosome loss strains were spotted on YPAD after selection on 5-FOA plates and measured for colony intensity. These values were then used to adjust the growth measurements for the other chromosome loss strains. Growth prediction of double disomies and double monosomies in high-throughput experiments was predicted by multiplying the growth of the two individual single disomies/monosomies. Epistasis scores were calculated by subtracting predicted growth from measured growth. This was done for each treatment individually. Determination of genetic interactions between disomic chromosomes at high drug concentrations was calculated from dilution series (Fig. 6A,C) by adding the measurements of the individual single disomies (DisA + DisB − 1). Growth greater/smaller than additive prediction +/− 30% were classified as a synergistic genetic interaction. The same method was used to identify genetic interactions between chromosome arms.

## Competition assay

Strains labeled with different fluorophores (*HTB2-mCherrs* or *HTB2*-mNeongreen) were first grown in 2 ml CSM lacking histidine, uracil, leucine, and lysine at 30 °C. Cell counts per culture were estimated using their optical densities. Co-cultures were normalized and mixed to obtain equal cell counts for each strain combination, making up the Day 0 time point. In total, 2 μl of the co-culture was transferred into the control condition (YPAD media with 0.6% DMSO) and 50 μl of the co-culture was transferred into YPAD with 30 μg/ml benomyl. Co-cultures were diluted into fresh media using the same regime every 24 h. In total, 800 μl of the culture was fixed with PFA to store samples for subsequent flow cytometry analysis. After the 3rd time point, cultures were struck out for single colonies on YPAD plates. The strain of origin for each colony was determined via fluorescence microscopy before karyotyping via qPCR.

## qPCR

For DNA extraction, cells were taken from plates and lysed in 20 μl of 0.02 M NaOH for 10 min at 100 °C. Lysates were centrifuged for

10 min on a mini-centrifuge (Thermo Fisher Scientific) to pellet cellular debris. Each 20 μl reaction contained 10 μl of Luna qPCR Mastermix (New England BioLabs Inc.), 1 μl of lysate, and 140 nM of each primer. Measurements were made on a RealPlex2 qPCR machine (Eppendorf). Cycling conditions were: activation/denaturation for 5 min at 95 °C followed by 40 cycles of 15 sec at 95 °C, 1 min at 60 °C with a subsequent fluorescence measurement. Melting curves were generated to determine primer specificity. *Ct* values were determined using the automatic thresholding of the RealPlex2 software. Chromosome copy numbers were calculated using a slightly modified ΔΔCt method (Schmittgen, 2001). The Ct values from duplicates were averaged and used to obtain the ΔCt, which was raised to the negative power of 2 to give the fold change. The ratio of fold change of the test strains to that of a wild-type strain was calculated to obtain the chromosome copy numbers. Karyotyping primers are in Appendix Table S4.

## Protein extraction and western blotting

For protein extraction, saturated ON cultures were diluted to an OD of 0.02 into YPAD. Cells were grown to an OD of ~0.6, normalized by OD, and harvested. Proteins were extracted by pelleting cells, resuspending them in 100 μl ddH$_2$O, adding 100 μl 0.2 M NaOH, and incubating them at room temperature for 5 min. Cells were pelleted and resuspended in 1× sample buffer before incubating them at 95 °C for 3 min. Samples were stored at −20 °C. Membranes were stained with 0.1% Ponceau S (Sigma-Aldrich) in 5% acetic acid. Total protein amount was detected with the ChemiDocMP imaging system (Bio-Rad). Membranes were washed for 30 min in Tris-buffered saline with 0.1% Tween (TBS-T) with three buffer exchanges, blocked for 1 h in 5% BSA (Sigma-Aldrich) or 5% skim milk (Gerbu Biotechnik GmbH) in TBS-T, depending on the primary antibody (see below). Primary antibodies diluted in their respective blocking buffers were incubated overnight at 4 °C on an orbital shaker. Membranes were washed for 30 min in TBS-T with three buffer exchanges before and after secondary antibody incubation. Horseradish peroxidase (HRP)-conjugated antimouse IgG secondary antibody (1:10,000; Cell Signaling Technology 7076) diluted in the respective blocking buffers was incubated for 1 h at room temperature. Chemiluminescence was developed using the Amersham ECL Prime detection reagent (Cytiva) and detected with the ChemiDocMP. Protein bands were quantified using ImageJ and normalized to the total protein amount quantified from the Ponceau S staining. The following primary antibodies were used: mouse anti-Pgk1 (in BSA and skim milk–TBST, respectively; 1:20,000; Thermo Fisher Scientific 22C5D8), anti-α-Tubulin (in skim milk–TBS; 1:20,000; mouse; Sigma-Aldrich T6074), anti-β-Tubulin (in BSA; 1:5.000; mouse; Developmental Studies Hybridoma Bank E7).

## Flow cytometry

Cultures were grown to log-phase, pelleted, and resuspended in 300 μl H$_2$O. 700 μl of 100% EtOH was added during vertexing, and cells were fixed for 1 h at room temperature. Cells were pelleted and resuspended in 50 mM sodium citrate, sonicated to disperse cell clumps, and treated with 250 μg/mL RNase A (Sigma-Aldrich, R6513) and 1 mg/mL Proteinase K (Sigma-Aldrich, P2308) overnight at 37 °C. Cells were resuspended in 50 mM sodium citrate

solution containing 1 µM SYTOX green (Thermo Fisher, 10768273). Co-cultures from the competition assay were pelleted and resuspended in 100 µl of 4% Paraformaldehyde (Sigma-Aldrich, 158127). Cells were pelleted and washed with 300 µl of KPO4/sorbitol and resuspended in 100 µl KPO4/sorbitol and stored in the fridge for up to 5 days. Samples were run on a BD LSRFortessa Cell Analyzer equipped with a 15-mW 488-nm laser and a 561-nm laser. Maximum count peaks for fluorescence intensities were calculated using the BD FACSDiva 8.0.1 software.

## Gene ontology enrichment analysis

For GO enrichment tests, the GOrilla gene ontology tool (http://cbl-gorilla.cs.technion.ac.il, accessed in September 2025) was used. The target gene list was composed of all genes identified as resistance genes, and the background list was all 6002 genes annotated in the *S. cerevisiae* genome. All lists of candidate genes were assembled from the Saccharomyces Genome Database, based on specific phenotype annotations. The list of Benomyl-related genes was based on the following classifications after benomyl treatment: resistance to chemicals: decreased; chromosome/plasmid maintenance: decreased; viability: decreased. 799 genes contain at least one of these terms. The list of CIN-related genes was assembled using the following GO terms: colony sectoring: increased; chromosome segregation: abnormal; chromosome/plasmid maintenance: decreased rate; chromosome/plasmid maintenance: abnormal; chromosome segregation (Clarke et al, 2023). In total, 718 different genes contain at least one of these terms. The list of tubulin/microtubule-related genes was based on: cytoplasmic microtubule organization; gammatubulin binding; microtubule cytoskeleton organization; microtubule depolymerization; microtubule nucleation; microtubule polymerization; microtubule-based process; mitotic spindle assembly; mitotic spindle elongation; tubulin binding; tubulin complex; tubulin complex assembly. In all, 43 genes contain at least one of these terms.

## RNA expression and protein abundance analysis

Data sets from Torres et al, 2007; Dephoure et al, 2014 and Muenzner et al, 2024 were used to analyze RNA and protein levels. Due to the high protein sequence homology between Tub1 and Tub3 (>90% identical), their protein levels were added together. Expression data from Muenzner et al were excluded from the analysis of benomyl resistance driver genes, as only four out of nine proteins were identified in their dataset.

## Statistical analysis

Statistical analysis was performed using GraphPad Prism software. Details for the statistical tests used in a particular experiment are reported in the figure legends.

# Data availability

This study includes no data deposited in external repositories.

The source data of this paper are collected in the following database record: biostudies:S-SCDT-10_1038-S44319-026-00767-8.

# Peer review information

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

## Acknowledgements

The authors thank the Campbell and Dammermann laboratories for helpful discussions and comments. We thank the Kraft lab and Kirk Anders for gifts of strains and plasmids, and Alexander Dammermann and Sean Montgomery for careful reading of the manuscript. We acknowledge the service of the FACS Facility. This work was supported by a Max Perutz Labs Fellowship to MSK and the Austrian Science Fund (FWF) grant F 8802-B to CSC.

## Author contributions

**M Sophie Koller**: Conceptualization; Data curation; Formal analysis; Writing—original draft; Writing—review and editing. **Claudia Himmelbauer**: Data curation; Formal analysis. **Sarah Fink**: Data curation; Formal analysis. **Madhwesh C Ravichandran**: Data curation; Formal analysis. **Christopher S Campbell**: Conceptualization; Formal analysis; Supervision; Funding acquisition; Writing—original draft; Writing—review and editing.

Source data underlying figure panels in this paper may have individual authorship assigned. Where available, figure panel/source data authorship is listed in the following database record: biostudies:S-SCDT-10_1038-S44319-026-00767-8.

## Disclosure and competing interests statement

The authors declare no competing interests.

# Expanded View Figures

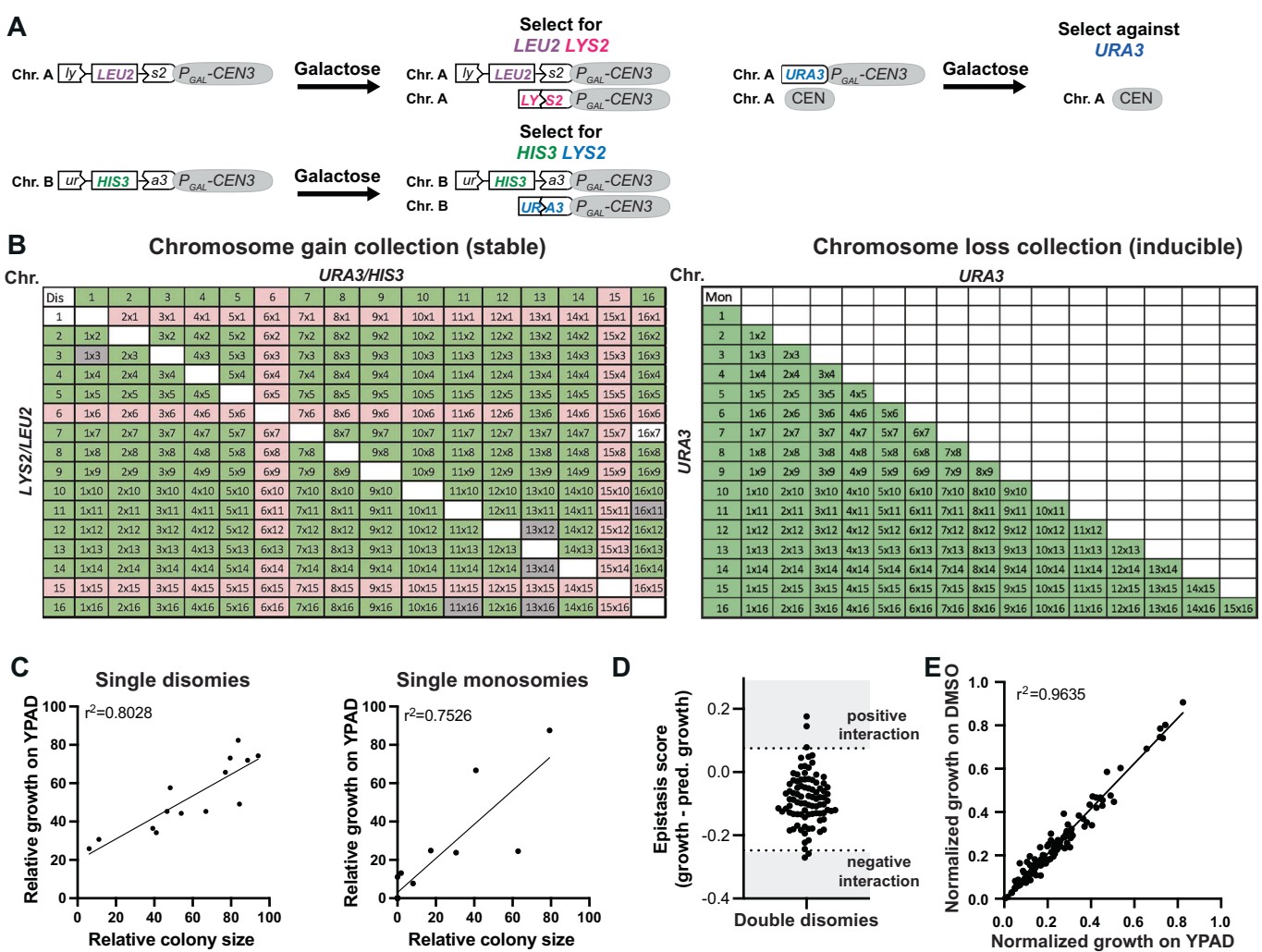

**Figure EV1. Engineering of the aneuploid yeast collections.**

(A) Schematic of the galactose-inducible system for engineering specific disomies and monosomies. Strains were grown in medium containing galactose to induce chromosome nondisjunction during mitosis. Monosomy of the desired chromosome was selected for on plates containing 5-Fluoroorotic Acid (5-FOA), selecting against the URA3 labeled chromosome. Disomy of the desired chromosome was selected for on minimal medium plates lacking leucine, and lysine, or histidine, and uracil, or all four amino acids. (B) Overview of the karyotypes represented in the chromosome gain collection (left) and the chromosome loss collection (right). Karyotypes highlighted in green are included in the collection, karyotypes highlighted in gray were excluded from the analysis due to diploidization and those highlighted in red are combinatorial lethal or the euploid parent strain had fitness defects. All possible single monosomies and monosomy combinations are inducible for chromosome loss, but not all are viable. (C) Correlation between growth of the single disomies (left) and monosomies (right) from two independent high-throughput spot assays to growth of the single disomies and monosomies from Beach et al, 2017. All strains were normalized to their respective WT controls (disomies: $r^2 = 0.749$, $P < 0.0001$, monosomies: $r^2 = 0.753$, $P < 0.0001$). $r^2$ values are from simple linear regression, and $P$ value are from F-tests. (D) Scatter plot of epistasis scores (measured growth - predicted growth) of the double disomies. Growth predictions are based on the product of the respective single disomies. Kolmogorov-Smirnov test for normality testing was used to test for a normal distribution (KS = 0.05935 and $P$ value > 0.1). Negative and positive interchromosomal interactions are defined by z-scores < −2 (−0.248) and >2 (0.075) and are indicated by dashed lines. (E) Correlation between quantified high-throughput spot assays from control plates (YPAD and YPAD + DMSO) ($r^2 = 0.9634$, $P < 0.0001$). $r^2$ values are from simple linear regression and $P$ value are from F-tests.

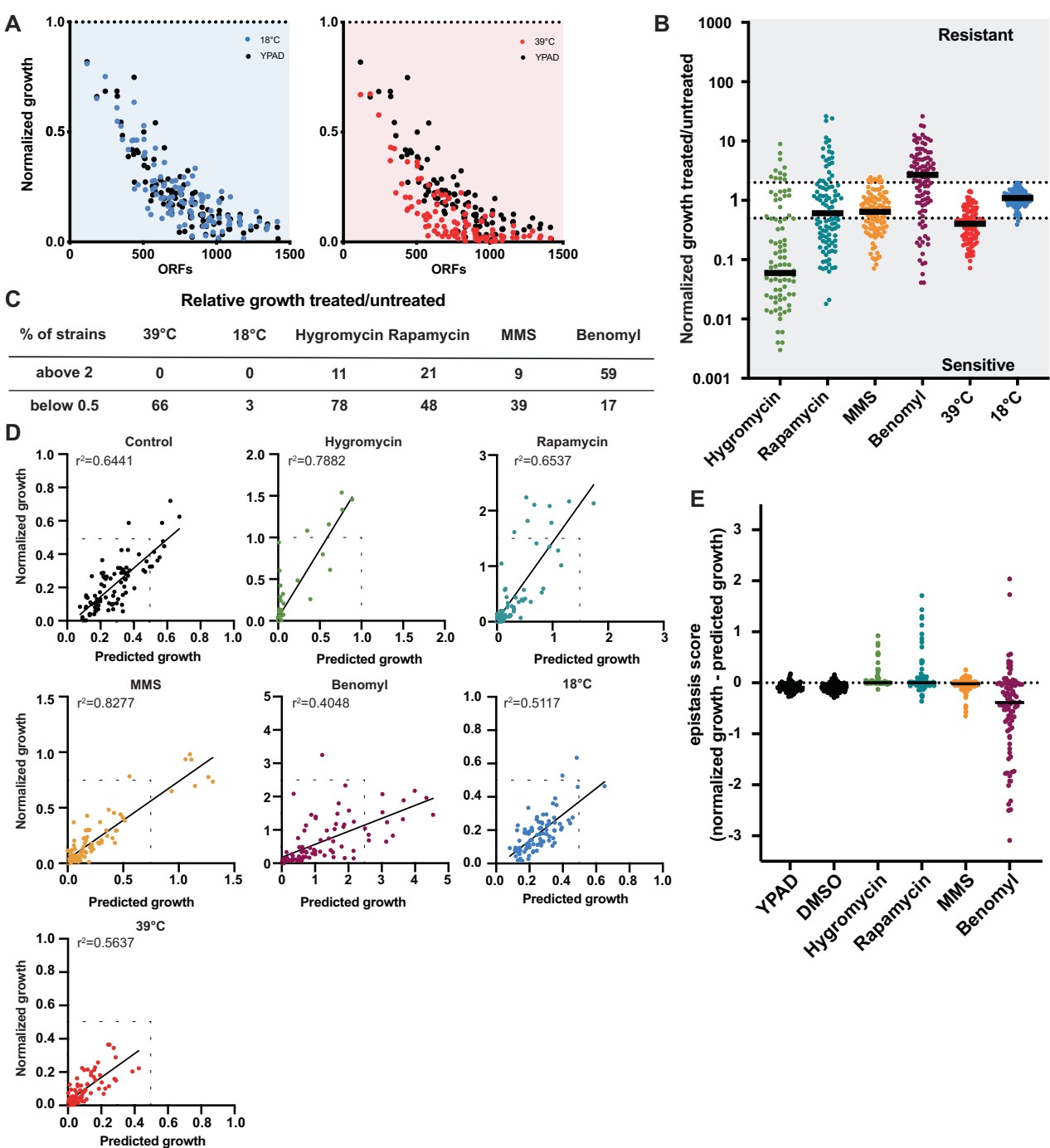

**Figure EV2. Genetic interactions between aneuploid chromosomes are more common under selective conditions.**

(A) Growth of strains from the chromosome gain collection. Control plates were quantified after 24 h (black) and temperature-shifted plates after 30 h (39 °C, red) and 48 h (18 °C, blue). All measurements were from two independent experiments and normalized to a haploid WT. (B) Relative growth of the chromosome gain collection under selective conditions normalized to both WT and untreated control plates. Relative growth >2 is counted as resistant, and relative growth <0.5 as sensitive. All measurements were from two independent experiments using 3–6 biological replicates. (C) Table of the percentage of strains that are resistant or sensitive to each treatment shown in (B). (D) Correlations between the growth of double disomies and predicted growth based on single disomies of two independent high-throughput growth assays. Assay conditions: YPAD averaged with YPAD + DMSO (black, $r^2 = 0.6441$, $P < 1.0 \times 10^{-15}$), YPAD + 40 μg/ml hygromycin (green, $r^2 = 0.7882$, $P < 1.0 \times 10^{-15}$), YPAD + 5 nM rapamycin (blue, $r^2 = 0.6537$, $P < 1.0 \times 10^{-15}$), YPAD + 0.02% MMS (orange, $r^2 = 0.8277$, $P < 1.0 \times 10^{-15}$), YPAD + 17.5 μg/ml benomyl (magenta, $r^2 = 0.4048$, $P = 2.08 \times 10^{-11}$), YPAD under cold stress (18 °C, blue, $r^2 = 0.5117$, $P = 3.0 \times 10^{-15}$) and YPAD under heat stress (39 °C, red, $r^2 = 0.5637$, $P < 1.0 \times 10^{-15}$). The dashed line indicates a perfect match between predicted growth and measured growth. $r^2$ values are from simple linear regression, and $P$ values are from F-tests. (E) Epistasis scores under different growth conditions. Negative scores indicate negative genetic interactions and positive scores indicate positive interactions. All measurements were from two independent experiments using 3–6 biological replicates.

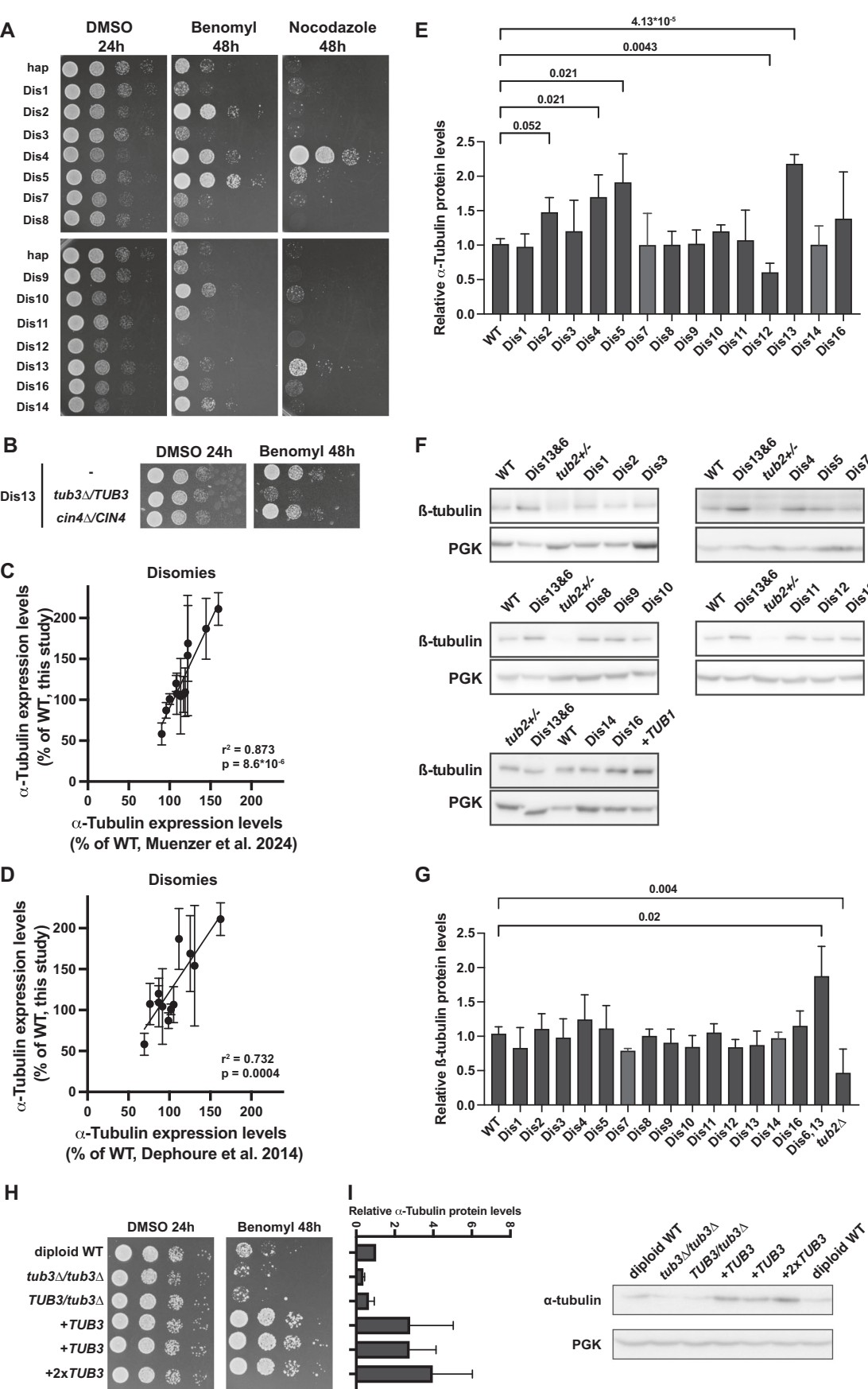

**Figure EV3. Aneuploidy can alter tubulin expression levels independently of tubulin copy numbers.**

(A) Tenfold serial dilution of all single disomies on agar plates containing DMSO (24 h), 25 µg/ml Benomyl, or 10 µg/ml Nocodazole. Disomies 4, 5, 10 and 13 are resistant to both drugs. (B) Tenfold serial dilution of disomy 13, disomy 13 tub3Δ/TUB3, and disomy 13 cin4Δ/CIN4 on YPAD + DMSO (24 h) and YPAD + 20 µg/ml benomyl (48 h) plates. Full serial dilution image including the WT control is in Appendix Fig. S3A. (C, D) Correlation between α-tubulin expression levels of the single disomies quantified by western blot (Fig. 3B,C, $n = 3$–4, biological) and α-tubulin expression levels of the Torres et al, 2007 single disomies quantified by mass spectrometry. Data from Muenzner et al, 2024 is shown in (C) ($P = 8.618 \times 10^{-6}$) and Dephoure et al, 2014 in (D) ($P = 0.0004$). $r^2$ and $P$ value are calculated from Pearson correlations. Means and standard deviations are shown. (E) Quantification of α-tubulin expression levels of the single disomies quantified by western blot (Fig. 3B, $n = 3$, biological). Mean and standard deviation are plotted. $P$ values are calculated using Brown–Forsythe and Welch ANOVA. (F) Western blot analysis of β-tubulin expression levels of all single disomies. The double disomy of chromosomes 6 and 13 and a tub2Δ/TUB2 diploid strain were used as controls for changes in β-tubulin expression levels. Pgk1 is shown as a loading control. (G) Quantification of ß-tubulin expression levels of the single disomies quantified by western blot ($n = 3$, biological). Mean and standard deviation are plotted. $P$ values are calculated using Brown–Forsythe and Welch ANOVA. (H) 10-fold dilution series of diploid strains with different TUB3 copy numbers on YPAD + DMSO (24 h) and YPAD + 15 µg/ml benomyl (48 h) plates. (I) Quantification of α-tubulin expression levels via western blot ($n = 3$, biological) for diploid strains with different TUB3 gene copy numbers. Means and standard deviations of three independent experiments are shown. A representative image from one of the blots is shown on the right.

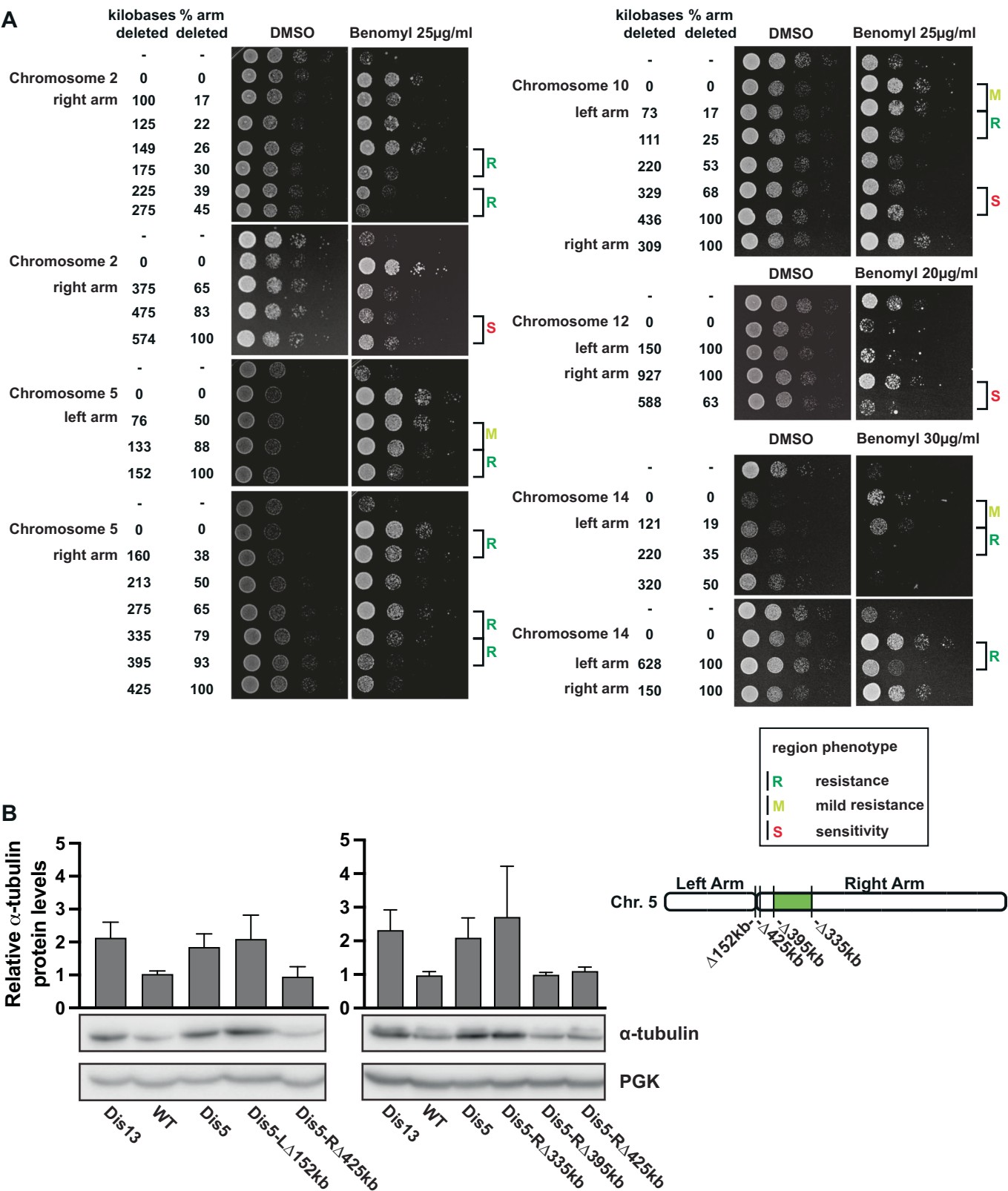

**Figure EV4. Multiple regions on aneuploid chromosomes contribute to benomyl resistance.**

(A) Tenfold serial dilutions of partial chromosome arm deletions for chromosomes 2, 4, 5, 10, 12, and 14. Control plates (YPAD + DMSO) were imaged after 24 h, and treated plates (YPAD + benomyl) after 48 h. (B) Quantification of α-tubulin expression levels of the single disomies quantified by western blot ($n = 3$, biological). Dis13 is a control for elevated α-tubulin levels. Mean and standard deviation are plotted. Quantified bands were normalized to the Ponceau-stained membrane and the euploid WT. The diagram on the right shows the location of the cuts on the right and left arms of chromosome 5. The region leading to α-tubulin overexpression is in green.

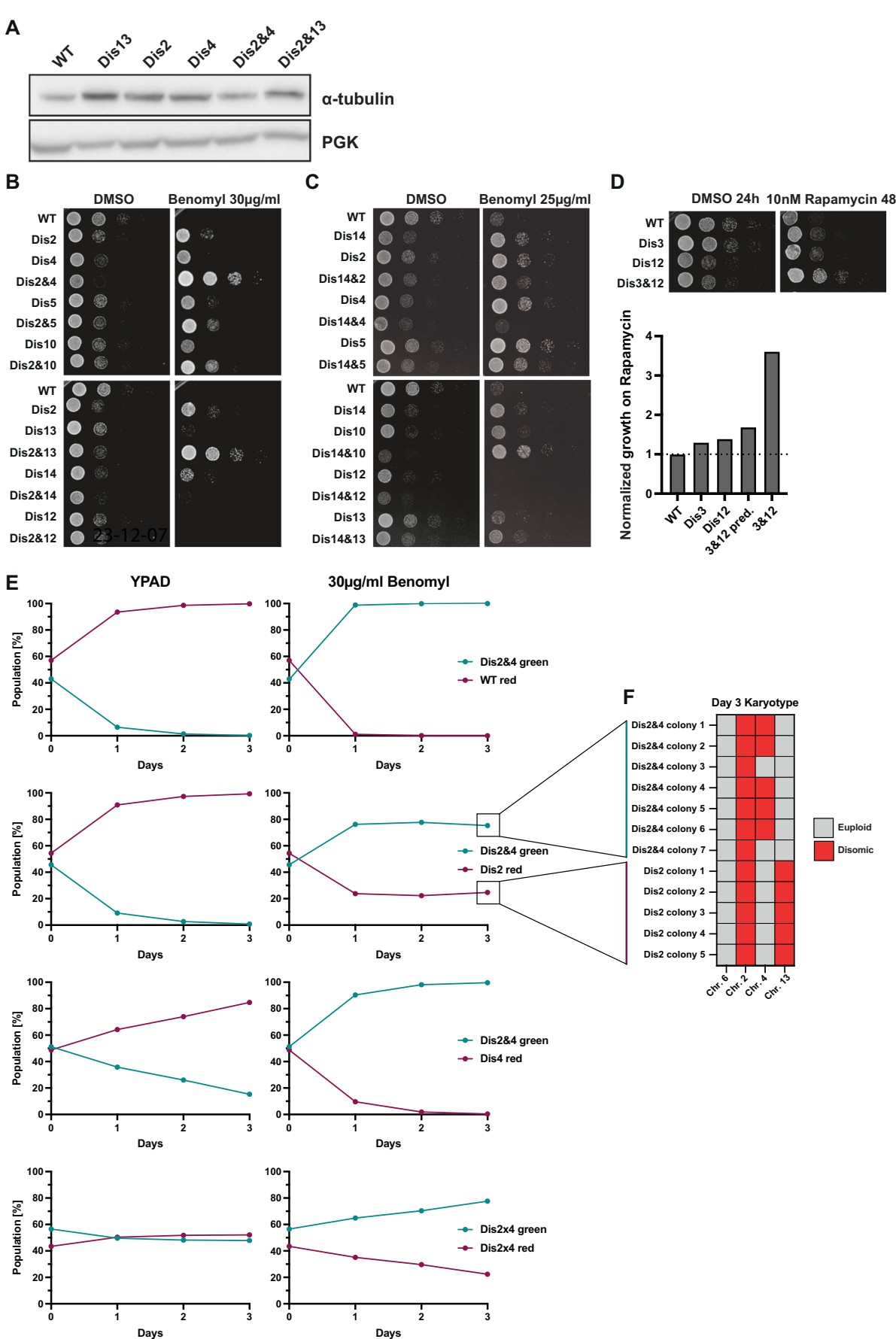

◀ **Figure EV5. Complex aneuploidy outcompetes single aneuploidies under drug selection.**

(**A**) Western blot analysis of α-tubulin expression levels for the indicated single and double disomies. (**B**) Tenfold serial dilutions of disomy combinations with chromosome 2. Control plates (YPAD + DMSO) were imaged after 24 h, and treated plates (YPAD + 30 μg/ml benomyl) after 48 h. (**C**) Tenfold serial dilutions of disomy combinations with chromosome 14. Control plates (YPAD + DMSO) were imaged after 24 h, and treated plates (YPAD + 25 μg/ml benomyl) after 48 h. (**D**) Tenfold serial dilutions and quantifications of disomy 3 and 12 on control (YPAD + DMSO) and treated (10 nM rapamycin) plates. All quantifications were normalized to a haploid WT. Combinatorial growth of chromosomes 3 and 12 double disomy was predicted by adding the phenotypes of the single disomies. (**E**) Liquid culture competition assay between different strains in control (YPAD + DMSO) and selective (YPAD + 30 μg/ml Benomyl) conditions. Cultures were diluted into fresh media every day, and samples were fixed with PFA and analyzed using flow cytometry. (**F**) qPCR measurements of chromosome copy numbers from single colonies from the 3rd day of the competition between Dis2 and Dis2&4 in selective (YPAD + 30 μg/ml Benomyl) conditions.

