## [Peer Review File · EMBO Reports]

Combinatorial effects of multiple genes contribute to beneficial aneuploidy phenotypes

Manuela Koller, Claudia Himmelbauer, Sarah Fink, Madhwesh Ravichandran, and Christopher Campbell

Corresponding author(s): Christopher Campbell (christopher.campbell@univie.ac.at)

Review Timeline:

Submission Date:	6th Oct 25
Editorial Decision:	31st Oct 25
Revision Received:	9th Jan 26
Editorial Decision:	6th Mar 26
Revision Received:	16th Mar 26
Accepted:	26th Mar 26

Editor: Esther Schnapp

Transaction Report:

Dear Prof. Campbell,

Thank you for the submission of your manuscript to EMBO reports. We have now received the full set of referee reports that is pasted below.

As you will see, all referees acknowledge that the findings are interesting. They mainly have suggestions for minor improvements, that should all be addressed. Referee 2 also suggests a few more experiments. I think point 3 of referee 2 is most interesting, and I don't think that you necessarily need to identify gene combinations for the publication of your study here, but please let me know what you think and we can discuss the exact revision requirements further, also in a video chat, if you like.

I would thus like to invite you to revise your manuscript with the understanding that the referee concerns must be fully addressed and their suggestions taken on board. Please address all referee concerns in a complete point-by-point response. Acceptance of the manuscript will depend on a positive outcome of a second round of review. It is EMBO reports policy to allow a single round of major revision only and acceptance or rejection of the manuscript will therefore depend on the completeness of your responses included in the next, final version of the manuscript.

We realize that it is difficult to revise to a specific deadline. In the interest of protecting the conceptual advance provided by the work, we recommend a revision within 3 months (31st Jan 2026). Please discuss the revision progress ahead of this time with the editor if you require more time to complete the revisions.

- 1) A data availability section providing access to data deposited in public databases is missing. If you have not deposited any data, please add a sentence to the data availability section that explains that.
- 2) Your manuscript contains statistics and error bars based on $n=2$. Please use scatter blots in these cases. No statistics should be calculated if $n=2$.

3) We replaced Supplementary Information with Expanded View (EV) Figures and Tables that are collapsible/expandable online. A maximum of 5 EV Figures can be typeset. EV Figures should be cited as 'Figure EV1, Figure EV2' etc... in the text and their respective legends should be included in the main text after the legends of regular figures.

5) a complete author checklist, which you can download from our author guidelines <https://www.embopress.org/page/journal/14693178/authorguide>. Please insert information in the checklist that is also reflected in the manuscript. The completed author checklist will also be part of the RPF.

6) Please note that all corresponding authors are required to supply an ORCID ID for their name upon submission of a revised manuscript (<https://orcid.org/>). Please find instructions on how to link your ORCID ID to your account in our manuscript tracking system in our Author guidelines <https://www.embopress.org/page/journal/14693178/authorguide#authorshipguidelines>

10) Regarding data quantification (see Figure Legends:

<https://www.embopress.org/page/journal/14693178/authorguide#figureformat>)

12) All Materials and Methods need to be described in the main text using our 'Structured Methods' format, which is required for all research articles. According to this format, the Methods section includes a separate Reagents and Tools Table file (listing key reagents, experimental models, software and relevant equipment and including their sources and relevant identifiers) and a Methods and Protocols section describing the methods using a step-by-step protocol format. The aim is to facilitate adoption of the methodologies across labs. More information on how to adhere to this format as well as a downloadable template (.docx) for the Reagents and Tools Table can be found in our author guidelines:

An example of a Method paper with Structured Methods can be found here: <https://www.embopress.org/doi/full/10.1038/s44320-024-00037-6#sec-4>

You are able to opt out of this by letting the editorial office know (emboreports@embo.org). If you do opt out, the Review

Process File link will point to the following statement: "No Review Process File is available with this article, as the authors have chosen not to make the review process public in this case."

I look forward to seeing a revised form of your manuscript when it is ready.

Referee #1:

Review of "Combinatorial effects of multiple genes contribute to beneficial aneuploidy phenotypes" from Koller et al.

Aneuploidy plays a well known role in adaptive evolution, originally observed in fungi and later confirmed in mammalian cells. Different data have shown that the compensatory effect can very rarely be attributed to changes in dosage of individual genes. However, no systematic efforts have been made to study how combinations of aneuploidies shape resistance to external stresses. The paper from Koller and collaborators aims at filling this gap with experiments performed in budding yeast. The authors have produced two collections of yeast strains carrying double disomies and chromosome losses. The collections have been treated with four different drugs, and growth has been analyzed and compared to WT cells.

The results show that double disomies have a wider array of gene combinations that provide resistance compared to single disomies. Interestingly, without selection double disomies show slight negative epistasis. However, under selection they can create new overexpression patterns that compensate for the stress caused by different drugs. Such patterns can give rise to positive epistasis, i.e. cannot be produced by single disomies.

The chromosome loss collection unfortunately was analyzed in much less depth. However, it explores a dimension of aneuploidy which has not been much investigated before and the collection poses the basis for future systematic analyses.

I find this work interesting and novel, and important for the field. I think the authors can make it sharper by clarifying the possible hypotheses they aim to test, and by presenting additional data that may reinforce the increased possibilities double disomic strains have to become resistant compared to single disomic strains.

Main comments

In the Introduction, the paper is presented as an investigation on interactions between aneuploid chromosomes, giving two alternative possibilities (lines 99-105). In the first, they mention the strong interaction between few genes, the example being the rescue operated by the gain of chromosome 13 on the lethality of chromosome 6 disomy. The second is the possibility that combinations of aneuploidies are the result of selective pressure. I do not see these as two alternative genetic interactions, both of them are true as we will discover reading the paper. Maybe the authors envisage two scenarios, interaction of few genes vs interaction of many genes? In any case, I find this part not so convincing. I think that it would be useful to appreciate the novelty of the results to explain more clearly the hypotheses that are going to be tested in the manuscript.

"While the majority of phenotypes are additive, the strongest resistance phenotypes of aneuploidies came from synergistic effects between different regions or chromosomes." I imagine what the authors refer to here are additive epistasis and positive epistasis (i.e., when the effect of combining alleles is what is expected from the individual alleles and when it is more than that). These terms are also used in the text. For clarity, I suggest to use only one terminology. I find that talking about positive epistasis/additive epistasis may be more precise and quantitative than about synergistic or additive effects and in any case these terms should be introduced explicitly.

- The production of the libraries needs to be commended. This was very hard work which provides new tools for the community of scientists interested in aneuploidy, resistance and chromosome segregation. Unfortunately, the loss collection cannot be selected for, but the authors have shown that this can be done on a large scale and produced the strains needed to do it.

- "The epistasis scores strongly adhered to a normal distribution (Kolmogorov-Smirnov test), indicating that most combinations do not deviate substantially from the expected growth rate (Fig S1D)." Here I have two comments, minor and major. In caption of Figure S1D it is reported that the result of the test is $KS = 0.05935$ and $p\text{-value} > 0.1$. The test $p > 0.1$ implies the distribution being normal? Second, the authors should give some references linking the distribution being normal and the fact that growth rates are as expected. This is not trivial to me. Aside from normality, Figure S1D indicates that the epistasis score seems to

have a negative average. Does this not imply negative epistasis, i.e. that double disomies perform on average worse than expected? If this is the case, it would further strengthen the authors' point: double disomies perform even worse than expected, and yet under selection...

Finally, I understand that growth is measured as colony size in Figure 1C at one time point. If so, this is not a growth rate but only growth. In general, it would be good to explain in the text how growth is measured. Related with this, in the manuscript there are plots that show the same things (for example, 1C and 6A) but carry different labels on the axes. This is confusing.

- "We conclude that synergistic interactions are rare in the absence of selective pressure". If we are talking about positive epistasis score, then I am not sure benomyl in percentage is much different (S1D vs S2E). Maybe it is, but the number of instances was not computed as far as I can see.

- The authors say: 'Overall, we conclude that the strongest drug resistance aneuploidy phenotypes typically come from the combined effects of multiple genes on different chromosomes', I guess here they refer to double disomies as opposed to single disomies. Similarly, in the Introduction they wrote that 'the strongest resistance phenotypes consistently came from combinations of aneuploid chromosomes, demonstrating the advantage of copy number changes in multiple genes'. For 'strong' they mean absolute growth? Absolute number?

In general, the comparison single/double disomies is a key point and it is not straightforward. There are more double than single disomies to begin with, and so the higher number may be expected when single disomies show resistance -- this is the case of benomyl (Figure 2C). Moreover, the sentence, while very appropriate for Rapamycin and Hygromycin, does not apply to MMS where the opposite is true: only single disomies grow better than WT.

However, I would say that even if not always true, the fact that multiple disomies can create new conditions that warrant resistance is important per se. One way to strengthen this point would be to show that there are combinations of disomies that show positive epistasis and resistance. Particularly interesting would be those that grow better than the WT while the predicted double disomies are not resistant. In this sense, one useful plot would have growth on the y-axis, and 4 genotypes on the x-axis: WT, disomy-1, disomy-2 and double disomy1+2. For the latter both predicted and measured growth should be plotted. The only point above the WT would be the measured growth. Are there such cases? How many are there? Extending the analysis to the other drugs may be useful (see last major comment).

More in general, all double disomies that show resistance and do not include only single disomies that alone give resistance help to make the point. For example, I find the combination of 2 and 13 very interesting (Figure 6): 13 on its own does not help, but it fosters the effect of 2 much more than expected. This remarks that the strong effect of combination of disomies observed in benomyl is not only driven by disomies that alone are already resistant. This is maybe the 'strong' effect the authors refer.

- The role of alpha-tubulin expression levels is new and very interesting. It also makes an interesting link to mammalian cells, where the modulation of isoforms of tubulin is one form of resistance to microtubule targeting drugs.

- I am not sure to understand the point of the last paragraph of the results section. The final sentence says: "We conclude that most aneuploidies display additive effects between genes, regions, and chromosomes. However, the strongest beneficial growth phenotypes often come from synergistic interactions." This is like saying that positive epistasis gives a stronger phenotype than additive epistasis. This is so by definition. Maybe this is the section where the authors could add some further analysis showing positive epistasis and resistance even with disomies that individually do not give resistance, for all 4 drugs.

Minor points

- "t still fails to explain why cells adapt through aneuploidy over other types of mutations that don't share its inherent downsides"
- the fitness trade-off of aneuploidy has been addressed quantitatively by Pompei and Lagomarsino (PNAS 2023). Here they show that the higher frequency of missegregation compared for example to point mutations plays a role. Maybe worth mentioning here.

- Figure 1D is in agreement with previous results, as reported by the authors. I may miss a point (is this piece of data relevant for the narrative since it describes a strong genetic interaction independently from selection?) but it does not strike me as a major new result.

- line 324, Fig S6B -> S5B

- line 326 chromosome arms (3/3) -> not clear what 3/3 refers to, the same applies to 2/8 line 333.

- "Since most of the resistance phenotypes were associated with the larger arms", I may have missed it, but this result was not discussed before.

- line 356 S5B -> S6B

- "For the two regions on chromosome 4 associated with the strongest benomyl resistance, we identified single gene deletions that decreased resistance to a similar extent as deleting the entire chromosomal region (Fig 5B and C)". I do not see any single gene deletion that does that in 5B and 5C. Deletion of SNQ2 in Figure S7A shows such an effect, but I do not understand why that is different from what I see in 5B.

- line 374, a reference on the prefoldin complex is needed.

- Figure S6: panel A,
chr2, an S between 125-149?
chr2, an R between 0-75?
chr14 left arm, an R between 0 and 628?

- Figure S7, disomy of chr2 does not give an improvement in the lane above npl4-null/NPL4

Referee #2:

Koller et al. show that beneficial aneuploidy phenotypes are largely combinatorial. Using near complete libraries of single and double chromosome gains and losses in budding yeast, they demonstrate that epistasis is rare in standard media yet becomes common and often positive under drug selection. The strongest drug resistance requires copy number changes in multiple chromosomes or multiple segments on the same chromosome, with clear examples in resistance to microtubule depolymerizers. For example, double disomies outperform singles under benomyl selection and link alpha tubulin abundance to benomyl resistance. While this study was primarily performed in yeast, it does address a highly fundamental question relevant to cancer biology that is very difficult to address in human cell lines.

The paper is elegant, comprehensive, and conceptually important. The experimental design allows a near-exhaustive dissection of combinatorial aneuploidy phenotypes, and the results convincingly establish that beneficial effects are often additive or synergistic across chromosomes and chromosomal segments. The work has clear implications for understanding how complex karyotypes shape drug resistance in cancer.

A few points could further strengthen the paper.

It might be helpful to clarify the causal role of alpha tubulin dosage in driving benomyl resistance. Since correlation does not imply mechanism, the authors could consider perturbing tubulin stoichiometry directly, for example by titrating TUB1 or TUB3 expression in both euploid and representative disomic strains, and measuring how resistance scales with total tubulin abundance or polymer mass. This would establish whether resistance arises from buffering, altered microtubule dynamics, or checkpoint remodeling.

The identification of multiple resistance regions on individual chromosomes is compelling. To move from regions to genes, the authors might consider reconstructing minimal combinations of candidate genes in a euploid background, using controlled overexpression or knock-in even for a single example. Testing for additivity versus synergy between these genes could directly validate the "combinatorial gene" hypothesis and show whether specific pathways (such as prefoldin, spindle checkpoint, or drug efflux) act cooperatively or independently.

The evolutionary and physiological consequences of complex karyotypes could also be explored further. For example, testing the relative stability and competitive fitness of single versus double disomies under fluctuating drug pressure would help assess whether combinatorial aneuploidies are transient adaptive states or stable solutions. This might also reveal whether some combinations promote further karyotypic evolution.

Finally, while the study focuses on yeast, a brief test of conservation, for instance, examining whether increased alpha tubulin dosage confers microtubule drug resistance in mammalian cells with engineered trisomies, would highlight translational relevance. Even a pilot experiment or literature comparison could help connect the mechanistic findings to cancer biology, where similar multi-segmental gains are common.

Overall, this is a rigorous and forward-thinking study that pushes the field toward a more quantitative and mechanistic understanding of complex aneuploidy. The suggestions above are meant to strengthen the link between correlation and causation and to underscore the broader relevance of the findings.

Referee #3:

The interesting manuscript "Combinatorial effects of multiple genes contribute to beneficial aneuploidy phenotypes" by Koller and colleagues from Chris Campbell's group addresses the question what are the advantages of aneuploidy, which is a frequent route to adaptation to stress conditions, and how the cells neutralize its detrimental consequences. Previously, it has been proposed that aneuploidy provides advantage through a) effect of one specific gene or, b) combinatorial consequences of simultaneous alteration multiple genes; at the same time the advantages of aneuploidy must outweigh its disadvantages. The authors generate a large collection of haploid yeast strains with many single and double disomies, as well as short-term monosomies induced in diploids for each possible chromosome and most combinations, and test their fitness in optimal growth settings as well as under various stress conditions. They also complement their analyses with elegant partial arm deletions to narrow down the regions contributing to specific phenotypes. Several interesting aspects were shown and previous suggestions were consolidated. The authors show that synergistic interactions are rather rare (most of the interactions were additive), and that there are also negative genetic interactions. They also show that the final phenotypes arise as a combination of both positive (increasing resistance) and negative effect (increasing sensitivity) of aneuploidy.

The manuscript is very clearly written and the work seems extremely well done and rather comprehensive. While there are no striking novel finding, this systematic analysis present robust findings and supports the previous observations that combinatorial effects of multiple genes underlie most of the positive impact of aneuploidy on cellular fitness. Therefore, I recommend the manuscript for publication and have only a few minor comments that should be clarified before acceptance.

Minor comments

1. The definition of Hygromycin B as ribosome inhibitor is rather unspecific. It should be better defined.
2. Could the authors speculate why there was a specific aneuploidy-based adaptation to hygromycin, rapamycin and benomyl treatment, but only very little for MMS treatment? This is rather interesting aspect.
3. Page 9, row 248: The authors should clarify following text:
"Although it has previously been reported that many disomies lead to benomyl sensitivity (Sheltzer et al. 2011; Torres et al. 2007), the resistance appears to outweigh the sensitivity for most chromosome combinations. Benomyl resistance is therefore not a "general" aneuploidy phenotype but is instead karyotype specific, despite being observed in the majority of double disomic strains."
This part is difficult to understand it should be probably rather: Benomyl SENSITIVITY is therefore not a "general" aneuploidy phenotype (because it has been previously claimed that aneuploid cells are generally sensitive to benomyl, not generally resistant).
4. The authors do excellent job with references, citing in an unbiased way most of relevant papers. Nevertheless, they might consider to expand references concerning the maintenance of aneuploid karyotypes over extended period of time (page 4, lane 78: e.g., Girish et al, 2023, Bokenkamp et al, 2025, Hintzen et al, 2024, and others).

We would like to thank the reviewers for their insightful and constructive comments. We feel that these suggestions have substantially improved the manuscript.

Referee #1:

Review of "Combinatorial effects of multiple genes contribute to beneficial aneuploidy phenotypes" from Koller et al.

Aneuploidy plays a well known role in adaptive evolution, originally observed in fungi and later confirmed in mammalian cells. Different data have shown that the compensatory effect can very rarely be attributed to changes in dosage of individual genes. However, no systematic efforts have been made to study how combinations of aneuploidies shape resistance to external stresses. The paper from Koller and collaborators aims at filling this gap with experiments performed in budding yeast. The authors have produced two collections of yeast strains carrying double disomies and chromosome losses. The collections have been treated with four different drugs, and growth has been analyzed and compared to WT cells.

The results show that double disomies have a wider array of gene combinations that provide resistance compared to single disomies. Interestingly, without selection double disomies show slight negative epistasis. However, under selection they can create new overexpression patterns that compensate for the stress caused by different drugs. Such patterns can give rise to positive epistasis, i.e. cannot be produced by single disomies.

The chromosome loss collection unfortunately was analyzed in much less depth. However, it explores a dimension of aneuploidy which has not been much investigated before and the collection poses the basis for future systematic analyses.

I find this work interesting and novel, and important for the field. I think the authors can make it sharper by clarifying the possible hypotheses they aim to test, and by presenting additional data that may reinforce the increased possibilities double disomic strains have to become resistant compared to single disomic strains.

Main comments

In the Introduction, the paper is presented as an investigation on interactions between aneuploid chromosomes, giving two alternative possibilities (lines 99-105). In the first, they mention the strong interaction between few genes, the example being the rescue operated by the gain of chromosome 13 on the lethality of chromosome 6 disomy. The second is the possibility that combinations of aneuploidies are the result of selective pressure. I do not see these as two alternative genetic interactions, both of them are true as we will discover reading the paper. Maybe the authors envisage two scenarios, interaction of few genes vs interaction of many genes? In any case, I find this part not so convincing. I think that it would be useful to appreciate the novelty of the results to explain more clearly the hypotheses that are going to be tested in the manuscript.

We agree that setting up two possibilities that are likely both true is confusing. We therefore deleted the second possibility and only mention that it is not currently clear how widespread genetic

interactions between chromosomes are and added an additional statement saying that it is also not known how such interaction could be influenced by selective conditions. We feel that this more clearly sets up the hypotheses.

"While the majority of phenotypes are additive, the strongest resistance phenotypes of aneuploidies came from synergistic effects between different regions or chromosomes." I imagine what the authors refer to here are additive epistasis and positive epistasis (i.e., when the effect of combining alleles is what is expected from the individual alleles and when it is more than that). These terms are also used in the text. For clarity, I suggest to use only one terminology. I find that talking about positive epistasis/additive epistasis may be more precise and quantitative than about synergistic or additive effects and in any case these terms should be introduced explicitly.

We agree and have now added more explicit definitions of the terms in the text (see lines 181-183). We have also attempted to clarify that "strong" phenotypes refer to degree of resistance not degree of epistasis, and now only use that descriptor for the level of resistance throughout the manuscript. We have still left in the term "synergistic" in certain locations to make the manuscript more easily understood by readers who are not classical geneticists. Thank you for helping us clarify our terminology.

- The production of the libraries needs to be commended. This was very hard work which provides new tools for the community of scientists interested in aneuploidy, resistance and chromosome segregation. Unfortunately, the loss collection cannot be selected for, but the authors have shown that this can be done on a large scale and produced the strains needed to do it.

Thank you.

- "The epistasis scores strongly adhered to a normal distribution (Kolmogorov-Smirnov test), indicating that most combinations do not deviate substantially from the expected growth rate (Fig S1D)." Here I have two comments, minor and major. In caption of Figure S1D it is reported that the result of the test is $KS = 0.05935$ and $p\text{-value} > 0.1$. The test $p > 0.1$ implies the distribution being normal?

Yes, a higher value indicates that the data are more likely to be normally distributed (the typical cutoff is 0.05).

Second, the authors should give some references linking the distribution being normal and the fact that growth rates are as expected. This is not trivial to me. Aside from normality, Figure S1D indicates that the epistasis score seems to have a negative average. Does this not imply negative epistasis, i.e. that double disomies perform on average worse than expected? If this is the case, it would further strengthen the authors' point: double disomies perform even worse than expected, and yet under selection...

Yes, the average is a bit on the negative side overall, suggesting that there may be a slight negative epistasis in general. However, these values are rather small and we felt they could reflect a slight general non-linearity in the growth measurements rather than actual genetic interactions. We therefore do not feel comfortable making a conclusive statement about it.

Finally, I understand that growth is measured as colony size in Figure 1C at one time point. If so, this

is not a growth rate but only growth. In general, it would be good to explain in the text how growth is measured. Related with this, in the manuscript there are plots that show the same things (for example, 1C and 6A) but carry different labels on the axes. This is confusing.

We have now added a short description of the measurement method to the results section in addition to the figure legends and methods section (see lines 142-143). We removed the word “rate” when it is not applicable. In addition, we have made the axis labeling more consistent.

- "We conclude that synergistic interactions are rare in the absence of selective pressure". If we are talking about positive epistasis score, then I am not sure benomyl in percentage is much different (S1D vs S2E). Maybe it is, but the number of instances was not computed as far as I can see.

Thanks for pointing this out. We meant to refer to either positive or negative interactions here (not just synergistic interactions) and have now clarified that in the text (lines 189-190).

- The authors say: 'Overall, we conclude that the strongest drug resistance aneuploidy phenotypes typically come from the combined effects of multiple genes on different chromosomes', I guess here they refer to double disomies as opposed to single disomies. Similarly, in the Introduction they wrote that 'the strongest resistance phenotypes consistently came from combinations of aneuploid chromosomes, demonstrating the advantage of copy number changes in multiple genes'. For 'strong' they mean absolute growth? Absolute number?

Indeed, “strong” here refers to absolute growth (relative to wild-type) as observed in Figure 6D for double disomies and in Figure 6A for single disomies. We removed any descriptions of genetic interactions as “strong” to avoid confusion.

In general, the comparison single/double disomies is a key point and it is not straightforward. There are more double than single disomies to begin with, and so the higher number may be expected when single disomies show resistance -- this is the case of benomyl (Figure 2C). Moreover, the sentence, while very appropriate for Rapamycin and Hygromycin, does not apply to MMS where the opposite is true: only single disomies grow better than WT.

Yes, this is true. There are indeed many more double than single disomies. We therefore avoided making any direct comparisons of the number of resistant strains in these two categories. Instead, we highlight that the highest resistance was often seen for chromosome combinations. It is true that this does not apply to MMS, where we did not observe substantial resistance for any of the aneuploid strains.

However, I would say that even if not always true, the fact that multiple disomies can create new conditions that warrant resistance is important per se. One way to strengthen this point would be to show that there are combinations of disomies that show positive epistasis and resistance. Particularly interesting would be those that grow better than the WT while the predicted double disomies are not resistant. In this sense, one useful plot would have growth on the y-axis, and 4 genotypes on the x-axis: WT, disomy-1, disomy-2 and double disomy1+2. For the latter both predicted and measured growth should be plotted. The only point above the WT would be the measured growth. Are there such cases? How many are there? Extending the analysis to the other drugs may be useful (see last major comment).

Indeed, interactions where the prediction would be a lack of resistance would be interesting and we did initially look for such examples. However, we did not find such cases. It seems that synergistic interactions come from weak phenotypes that are then much stronger when combined. There are some examples where only one chromosome shows resistance, which is then higher when combined with a non-resistant chromosome. This includes the interaction that the authors point out below for chromosomes 2 and 13 at the higher benomyl concentration. In the end, we found it most helpful to highlight the cases with the highest levels of overall resistance.

More in general, all double disomies that show resistance and do not include only single disomies that alone give resistance help to make the point. For example, I find the combination of 2 and 13 very interesting (Figure 6): 13 on its own does not help, but it fosters the effect of 2 much more than expected. This remarks that the strong effect of combination of disomies observed in benomyl is not only driven by disomies that alone are already resistant. This is maybe the 'strong' effect the authors refer.

Chromosome 13 is tricky, as its resistance is highly concentration-dependent. It provides resistance at lower concentrations of benomyl (Figure 3A), but not so much at higher concentrations (Figure 6B). As the reviewer point out, at the higher benomyl concentration, chromosome 13 disomy does not provide any resistance on its own, yet greatly enhances the resistance when combined with chromosome 2 disomy. We have now emphasized this intriguing example of genetic interaction in the text. Thank you for the suggestion. (see lines 452-455). As noted above, "strong" was not meant to refer to the degree of genetic interaction.

- The role of alpha-tubulin expression levels is new and very interesting. It also makes an interesting link to mammalian cells, where the modulation of isoforms of tubulin is one form of resistance to microtubule targeting drugs.

Agreed. Although we did not find a correlation between the copy number of alpha-tubulin isoforms and their expression levels on Depmap (see response to reviewer 2 below), it is still possible that there are meaningful contributions to tubulin expression levels that are masked by the variability in expression due to other factors in the different cell lines.

- I am not sure to understand the point of the last paragraph of the results section. The final sentence says:

"We conclude that most aneuploidies display additive effects between genes, regions, and chromosomes. However, the strongest beneficial growth phenotypes often come from synergistic interactions." This is like saying that positive epistasis gives a stronger phenotype than additive epistasis. This is so by definition. Maybe this is the section where the authors could add some further analysis showing positive epistasis and resistance even with disomies that individually do not give resistance, for all 4 drugs.

Hopefully we have already clarified this with our responses above. "Strongest beneficial growth phenotypes" here refer to the highest measurements of absolute growth, not highest degree of epistasis.

Minor points

- "t still fails to explain why cells adapt through aneuploidy over other types of mutations that don't share its inherent downsides" - the fitness trade-off of aneuploidy has been addressed quantitatively by Pompei and Lagomarsino (PNAS 2023). Here they show that the higher frequency of missegregation compared for example to point mutations plays a role. Maybe worth mentioning here.

We have added that reference. Thank you.

- Figure 1D is in agreement with previous results, as reported by the authors. I may miss a point (is this piece of data relevant for the narrative since it describes a strong genetic interaction independently from selection?) but it does not strike me as a major new result.

Figure 1D shows a genetic interaction between chromosomes 13 and 6 for chromosome loss. This has previously only been demonstrated for chromosome gain.

- line 324, Fig S6B -> S5B

Fixed. Thanks.

- line 326 chromosome arms (3/3) -> not clear what 3/3 refers to, the same applies to 2/8 line 333.

This was in reference to Fig 4E. We have now simply removed them to avoid confusion.

- "Since most of the resistance phenotypes were associated with the larger arms", I may have missed it, but this result was not discussed before.

Yes, this was mentioned earlier in the manuscript: "For chromosomes 2, 4, 10, and 14, disomy of the larger arm was sufficient to fully recapitulate the benomyl resistance." (line 315-317)

- line 356 S5B -> S6B

Fixed. Thanks.

- "For the two regions on chromosome 4 associated with the strongest benomyl resistance, we identified single gene deletions that decreased resistance to a similar extent as deleting the entire chromosomal region (Fig 5B and C)". I do not see any single gene deletion that does that in 5B and 5C. Deletion of SNQ2 in Figure S7A shows such an effect, but I do not understand why that is different from what I see in 5B.

The comparisons in Figure 5B are unfortunately not straight forward. Importantly, we are looking for phenotypes that match what is observed for a single region, not the whole arm. For the SWR1+/- deletion, it should be compared to the 460kb deletion. These are similar in growth on the benomyl plate. For the SNQ2+/- deletion, we do not have a deletion of just the region that contains the gene. In this case, we can compare the 1082kb deletion to the 1032kb plus SNQ2+/- deletion. These are also quite similar.

The difference in phenotype for the SNQ2+/- deletion between 5B and S7A can be attributed to variability in experiments conducted at different times. The effect of the benomyl plate on WT noticeably different between two plates, indicating that the potency of the benomyl was different

despite our best efforts to maintain consistency over time.

- line 374, a reference on the prefoldin complex is needed.

We have added the following reference: PMID: 9630229.

- Figure S6: panel A,
chr2, an S between 125-149?
chr2, an R between 0-75?

Yes, these regions seem to indicate sensitivity and resistance in the provided example. Unfortunately, we did not find these growth differences to be reproducible enough to feel comfortable designating them regions of sensitivity or resistance.

chr14 left arm, an R between 0 and 628?

Yes, the chromosome 14 left arm 0-628kb region shows strong resistance and is the full left arm deletion. We added an R to the figure.

- Figure S7, disomy of chr2 does not give an improvement in the lane above npl4-null/NPL4.

Well observed. Disomy 2 did not show the characteristic resistance in one of the images. Thank you for pointing this out. We have now repeated the experiment and replaced the image.

Referee #2:

Koller et al. show that beneficial aneuploidy phenotypes are largely combinatorial. Using near complete libraries of single and double chromosome gains and losses in budding yeast, they demonstrate that epistasis is rare in standard media yet becomes common and often positive under drug selection. The strongest drug resistance requires copy number changes in multiple chromosomes or multiple segments on the same chromosome, with clear examples in resistance to microtubule depolymerizers. For example, double disomies outperform singles under benomyl selection and link alpha tubulin abundance to benomyl resistance. While this study was primarily performed in yeast, it does address a highly fundamental question relevant to cancer biology that is very difficult to address in human cell lines.

The paper is elegant, comprehensive, and conceptually important. The experimental design allows a near-exhaustive dissection of combinatorial aneuploidy phenotypes, and the results convincingly establish that beneficial effects are often additive or synergistic across chromosomes and chromosomal segments. The work has clear implications for understanding how complex karyotypes shape drug resistance in cancer.

A few points could further strengthen the paper.

It might be helpful to clarify the causal role of alpha tubulin dosage in driving benomyl resistance. Since correlation does not imply mechanism, the authors could consider perturbing tubulin stoichiometry directly, for example by titrating TUB1 or TUB3 expression in both euploid and representative disomic strains, and measuring how resistance scales with total tubulin abundance or

polymer mass. This would establish whether resistance arises from buffering, altered microtubule dynamics, or checkpoint remodeling.

Good point. Directly altering alpha tubulin levels would be more convincing than the correlations that we currently show. We therefore engineered diploid yeast with varying copy numbers of the *TUB3* gene (new Figure EV3 H, I). We see that decreased *TUB3* copy number substantially decreases benomyl resistance and that increased *TUB3* copy number substantially increases resistance.

The identification of multiple resistance regions on individual chromosomes is compelling. To move from regions to genes, the authors might consider reconstructing minimal combinations of candidate genes in a euploid background, using controlled overexpression or knock-in even for a single example. Testing for additivity versus synergy between these genes could directly validate the "combinatorial gene" hypothesis and show whether specific pathways (such as prefoldin, spindle checkpoint, or drug efflux) act cooperatively or independently.

We agree that this would have strengthened the manuscript and we already tried it previously, focusing on chromosome 5 where we identified the highest number of genes that decrease resistance when heterozygously deleted in the disomic strain. We engineered integration plasmids with the genes of interest, native promoter, and 3' UTR. However, in contrast to the *TUB3* containing plasmid, none of the genes on chromosome 5 showed substantial resistance on their own or in combinations of two genes. Although this agrees with our conclusion that chromosome 5 resistance result from the synergistic effects of many genes, we do not feel that the inclusion of such negative data would substantially improve the manuscript.

The evolutionary and physiological consequences of complex karyotypes could also be explored further. For example, testing the relative stability and competitive fitness of single versus double disomies under fluctuating drug pressure would help assess whether combinatorial aneuploidies are transient adaptive states or stable solutions. This might also reveal whether some combinations promote further karyotypic evolution.

Thanks to this suggestion, we have now conducted competition assays between strains with disomy of chromosomes 2 and 4 and either the single disomies or the euploid strain (new figure EV5 E, F). As expected, the double disomic strain was always outcompeted under non-selective conditions. With the addition of benomyl, the double disomic strain initially outcompeted all of the other strains. Intriguingly, the disomy 2 strain was not fully outcompeted and eventually stabilized. qPCR analysis of these strains showed that they had now gained an extra copy of chromosome 13 in addition to the preexisting chromosome 2 disomy. This agrees with our observation that the 2&4 and 2&13 disomies have similarly high levels of benomyl resistance (Figure 6D). We conclude that the double 2&4 disomy provides a strong competitive advantage in the short term. In the longer term, already having one beneficial aneuploidy can allow for more beneficial complex karyotypes to form before they are fully outcompeted.

Finally, while the study focuses on yeast, a brief test of conservation, for instance, examining whether increased alpha tubulin dosage confers microtubule drug resistance in mammalian cells with engineered trisomies, would highlight translational relevance. Even a pilot experiment or literature comparison could help connect the mechanistic findings to cancer biology, where similar multi-segmental gains are common.

Translating our findings about tubulin copy number to human cells is challenging for a number of reasons. First, both alpha and beta Tubulin have multiple isoform in humans whose genes are largely scattered across the genome, so individual aneuploidies may have less of an effect on tubulin levels. Second, the absolute levels of alpha and beta tubulin may be less important than the ratios in expression between the various isoforms (PMID: 28677634). Third, there are additional mechanisms that have been reported to keep tubulin protein levels stable through posttranscriptional regulation (PMID: 8978820). Finally, tubulin expression varies greatly in different cell types. Therefore, to properly address this question it would be necessary to engineer aneuploidies for a large variety of chromosomes in human cells and measure gene expression, protein level, and resistance to spindle poisons. Such a study would be quite interesting but unfortunately falls outside the scope of the 3-month revision period. As an alternative, we have explored the Depmap database to determine if tubulin copy number correlates with resistance to benomyl. However, changes in copy number of alpha tubulin genes does not even correlate with alpha tubulin protein levels in these cell lines either when analyzing the isoforms individually or in combination (see reviewer Figure 1). This is likely due to the high variability in tubulin gene expression across these cell lines for reasons such as those stated above.

Overall, this is a rigorous and forward-thinking study that pushes the field toward a more quantitative and mechanistic understanding of complex aneuploidy. The suggestions above are meant to strengthen the link between correlation and causation and to underscore the broader relevance of the findings.

Referee #3:

The interesting manuscript "Combinatorial effects of multiple genes contribute to beneficial aneuploidy phenotypes" by Koller and colleagues from Chris Campbell's group addresses the question what are the advantages of aneuploidy, which is a frequent route to adaptation to stress conditions, and how the cells neutralize its detrimental consequences. Previously, it has been proposed that aneuploidy provides advantage through a) effect of one specific gene or, b) combinatorial consequences of simultaneous alteration multiple genes; at the same time the advantages of aneuploidy must outweigh its disadvantages. The authors generate a large collection of haploid yeast strains with many single and double disomies, as well as short-term monosomies induced in diploids for each possible chromosome and most combinations, and test their fitness in optimal growth settings as well as under various stress conditions. They also complement their analyses with elegant partial arm deletions to narrow down the regions contributing to specific phenotypes. Several interesting aspects were shown and previous suggestions were consolidated. The authors show that synergistic interactions are rather rare (most of the interactions were additive), and that there are also negative genetic interactions. They also show that the final phenotypes arise as a combination of both positive (increasing resistance) and negative effect (increasing sensitivity) of aneuploidy.

The manuscript is very clearly written and the work seems extremely well done and rather comprehensive. While there are no striking novel finding, this systematic analysis present robust

findings and supports the previous observations that combinatorial effects of multiple genes underlie most of the positive impact of aneuploidy on cellular fitness. Therefore, I recommend the manuscript for publication and have only a few minor comments that should be clarified before acceptance.

Minor comments

1. The definition of Hygromycin B as ribosome inhibitor is rather unspecific. It should be better defined.

We have now changed this to “inhibitor of ribosomal polypeptide synthesis”, as hygromycin B inhibits translocation of the nascent polypeptide through the ribosome.

2. Could the authors speculate why there was a specific aneuploidy-based adaptation to hygromycin, rapamycin and benomyl treatment, but only very little for MMS treatment? This is rather interesting aspect.

Our answer here is indeed speculative. One possibility is that the target of MMS is DNA instead of a protein (or ribonucleoprotein in the case of hygromycin), and therefore acquiring resistance due to changes in gene copy number is not as straight forward. For rapamycin and benomyl, one of the key mechanisms of resistance is changing the expression levels of the drug target (Tor1 and tubulin, respectively). This is not possible for MMS treatment.

3. Page 9, row 248: The authors should clarify following text:

"Although it has previously been reported that many disomies lead to benomyl sensitivity (Sheltzer et al. 2011; Torres et al. 2007), the resistance appears to outweigh the sensitivity for most chromosome combinations. Benomyl resistance is therefore not a "general" aneuploidy phenotype but is instead karyotype specific, despite being observed in the majority of double disomic strains."

This part is difficult to understand it should be probably rather: Benomyl SENSITIVITY is therefore not a "general" aneuploidy phenotype (because it has been previously claimed that aneuploid cells are generally sensitive to benomyl, not generally resistant).

We understand the confusion. It has been previously claimed that benomyl sensitivity is a general aneuploidy phenotype. We, on the other hand, observe resistance in the majority of our aneuploid strains (Figure S2B). We have now modified the text to say that neither resistance nor sensitivity is a general aneuploidy phenotype (line 250-251). We hope this clarifies things.

4. The authors do excellent job with references, citing in an unbiased way most of relevant papers. Nevertheless, they might consider to expand references concerning the maintenance of aneuploid karyotypes over extended period of time (page 4, line 78: e.g., Girish et al, 2023, Bokenkamp et al, 2025, Hintzen et al, 2024, and others).

These references have now been added. Thank you.

Reviewer Figure 1. Alpha tubulin copy number and protein levels do not correlate in human cancer cell lines. Analyses from depmap portal (depmap.org) for (A) 5 different alpha tubulin isotypes or (B) the average of the three most highly expressed isotypes (TUBA1B, TUBA1C, and TUBA4A).

Dear Prof. Campbell,

Thank you for the submission of your revised manuscript. We have now received the enclosed reports from referee 1, who was asked to assess it. Unfortunately, referee 2 was not available to re-review your ms and referee 1 has therefore also assessed your response to referee 2's comments. I am happy to say that referee 1 supports the publication of your revised ms, and only a few editorial requests remain to be addressed before we can proceed with the official acceptance of your manuscript:

- Please correct the conflict of interest subheading to "Disclosure and Competing Interests Statement" and place it after the Acknowledgement section.
- Please remove the author credits from the ms file. All credits need to be entered during online ms submission.
- When submitting the final ms online, please add the Max Perutz Labs Fellowship as a separate Funder in our online submission system.
- Supplemental Fig. S4A, B and Supplemental Fig. S7A are incorrect callouts, and need to be updated to the correct nomenclature.
- The figures in the Appendix file are missing the word "Appendix" in their title and legends throughout the file. The correct name is Appendix Figure S1, etc.
- In the Reagents & Tools table there are some incorrect callouts that need to be corrected since these items do not exist - Table EV 3, Table EV 5.
- Material and methods should be just Methods.
- Our routine image analysis of to be accepted ms detected a potential partial cell reuse between Figure EV3B and Appendix Fig S3A that is not listed in the Figure legends. Can you please explain/clarify?

Figure Legends - Comments

- Please define the annotated p values ****/***/**/* as well as provide the exact p-values for the same in the legend of figure 4D, 5A, B as appropriate and reasonable.
- Please note that the exact p values are not provided in the legends of figures 1C, 3C; EV3 C, E, please provide exact values as reasonable.
- Please note that information related to n is missing in the legends of figures EV2 B, E; EV3 C
- Please note that the error bars are not defined in the legend of figure EV3 C.

EMBO press papers are accompanied online by A) a short (1-2 sentences) summary of the findings and their significance, B) 2-3 bullet points highlighting key results and C) a synopsis image that is exactly 550 pixels wide and 200-600 pixels high (the height is variable). The synopsis image should provide a sketch of the major findings, like a graphical abstract. Please note that text needs to be readable at the final size. Please send us this information along with the final manuscript.

Referee #1:

The authors addressed my concerns satisfactorily. The modified manuscript is improved, and key concepts have been clarified. It includes new additional competition experiments that further reinforce the authors' conclusions. I recommend publication of the article in EMBO Reports.

Referee 1's comments on referee 2's report:

- First, it needs to be noticed that Reviewer 2 was very supportive of the study to begin with - the points raised were meant to further improve the study.
- In particular, Reviewer 2 asked to: (i) turn correlative observations into causative explanations (tubulin levels and resistance; competition between disomies and monosomies; expression of minimal gene combinations independently from ploidy changes); (ii) widen the relevance of the results to mammalian cells.
- The causative effect was clearly observed with the modulation of Tub3 levels and also with the competition experiments. The expression of minimal gene combination was unsuccessful. Maybe choosing the chromosome with the highest number of contributing genes at the end was not a good idea given that all of them may turn out to be needed. However, I think that the main observation (fitness improved by the interaction of many genes) emerges quite clearly by the segmental deletions anyway.
- The generality issue was not addressed. However, I agree that extending to mammals the observed connection between aneuploidy and resistance to tubulin drugs is not trivial. Due to the complexity of tubulin genes in mammals as opposed to yeast and the many layers of control of the different isoforms, I do not think this point could be proven with any simple experiment. Hence, all in all I think the suggestions of Reviewer 2 improved the paper, as they prompted new experiments that show the causative effect of Tub3 levels and confirm the doubles disomies being more efficient than single ones.

The authors have addressed all minor editorial requests.

Prof. Christopher Campbell
Vienna Biocenter (VBC)
MAX F. PERUTZ LABORATORIES
Dr. Bohr-Gasse 9
Vienna, Vienna 1030
Austria

Dear Prof. Campbell,

I am very pleased to accept your manuscript for publication in the next available issue of EMBO reports. Thank you for your contribution to our journal.

You may qualify for financial assistance for your publication charges - either via a Springer Nature fully open access agreement or an EMBO initiative. Check your eligibility: <https://link.springer.com/journal/44319/how-to-publish-with-us>

>>> Please note that it is EMBO Reports policy for the transcript of the editorial process (containing referee reports and your response letter) to be published as an online supplement to each paper. If you do NOT want this, you will need to inform the Editorial Office via email immediately. More information is available here: <https://link.springer.com/partners/embo-press/editorial-policies#Peer%20review>